# Modeling the spatiotemporal spread of beneficial alleles using ancient genomes

**Rasa A Muktupavela[1]\*, Martin Petr[1], Laure Ségurel[2], Thorfinn Korneliussen[1], John Novembre[3], Fernando Racimo[1]**

[1]Lundbeck GeoGenetics Centre, GLOBE Institute, Faculty of Health, Copenhagen, Denmark; [2]UMR5558 Biométrie et Biologie Evolutive, CNRS - Université Lyon 1, Villeurbanne, France; [3]Department of Human Genetics, University of Chicago, Chicago, United States

**Abstract** Ancient genome sequencing technologies now provide the opportunity to study natural selection in unprecedented detail. Rather than making inferences from indirect footprints left by selection in present-day genomes, we can directly observe whether a given allele was present or absent in a particular region of the world at almost any period of human history within the last 10,000 years. Methods for studying selection using ancient genomes often rely on partitioning individuals into discrete time periods or regions of the world. However, a complete understanding of natural selection requires more nuanced statistical methods which can explicitly model allele frequency changes in a continuum across space and time. Here we introduce a method for inferring the spread of a beneficial allele across a landscape using two-dimensional partial differential equations. Unlike previous approaches, our framework can handle time-stamped ancient samples, as well as genotype likelihoods and pseudohaploid sequences from low-coverage genomes. We apply the method to a panel of published ancient West Eurasian genomes to produce dynamic maps showcasing the inferred spread of candidate beneficial alleles over time and space. We also provide estimates for the strength of selection and diffusion rate for each of these alleles. Finally, we highlight possible avenues of improvement for accurately tracing the spread of beneficial alleles in more complex scenarios.

**\*For correspondence:**
rasa.muktupavela@gmail.com

**Competing interest:** The authors declare that no competing interests exist.

## Editor's evaluation

This is an important manuscript that presents an elegant framework to infer the dynamics of beneficial alleles over time and space. The authors present a new method and show, convincingly, its utility and great potential to reconstruct the evolutionary history of beneficial alleles. The method is also applied to loci that likely mediated human genetic adaptations, contributing to our understanding of human recent evolution. The work will be of broad interest to evolutionary biologists who seek to understand the dynamics of beneficial mutations in populations.

## Introduction

Understanding the dynamics of the spread of a beneficial allele through a population is one of the fundamental problems in population genetics (*Ewens, 2012*). We are often interested in knowing the location where an allele first arose and the way in which it spread through a population, but this is often unknown, particularly in natural, non-experimental settings where genetic sampling is scarce and uneven.

Patterns of genetic variation can be used to estimate how strongly natural selection has affected the trajectory of an allele and to fit the parameters of the selection process. The problem of estimating

**eLife digest** Analyzing the genomes of our ancient ancestors can reveal how certain traits spread through the human population over the course of evolution. Mutations that make individuals better equipped to survive their environment are more likely to be passed on to the next generation and become more common. For example, a genetic variant that enables adult people to digest sugars in dairy products has become more common in humans over time. Yet evolution does not only happen across time: it transverses space as well.

Modeling the geographic spread of such genetic mutations is challenging using existing methods. To overcome this, Muktupavela et al. developed a new computational method that uses modern and ancient human genomes to study the evolution of specific genetic variants across space and time. The tool can determine where certain variants first emerged, how quickly they spread across geographic areas, and how rapidly they became prevalent in human populations.

Muktupavela et al. applied their new method, which was based on a previously published framework, to track the spread of two common genetic variations that have previously been reported to be subject to natural selection: one that allows adult humans to digest dairy products, and another associated with skin pigmentation. They found that the mutation that enabled dairy consumption originated around what is now southwestern Russia or eastern Ukraine. The variation then spread westward, becoming increasingly more common over the course of the Holocene.

The mutation related to skin pigmentation emerged further south than the dairy-related variation, and then also spread westward. Massive human migrations during the Neolithic and Bronze Age eras may have helped disperse both variants.

The model developed by Muktupavela et al. could help scientists track the geographic spread of other genetic variants in human populations, as well as provide new insights into how humans adapt to changing environmental conditions. Incorporating major events into the model, like mass migrations or glacial retreats, may lead to even more insights.

the age of a beneficial allele, for example, has yielded a rich methodological literature (**Slatkin and Rannala, 2000**), and recent methods have exploited fine-scale haplotype information to produce highly accurate age estimates (**Mathieson and McVean, 2014**; **Platt et al., 2019**; **Albers and McVean, 2020**). In contrast, efforts to infer the geographic origins of beneficial mutations are scarcer. These include **Novembre et al., 2005**, who developed a maximum likelihood method to model the origin and spread of a beneficial mutation and applied it to the *CCR5-Δ32* allele, which was, at the time, considered to have been under positive selection (**Stephens et al., 1998**; **Sabeti et al., 2005**; **Novembre and Han, 2012**). Similarly, **Itan et al., 2009** developed an approximate Bayesian computation (ABC) approach using demic simulations, in order to find the geographic and temporal origins of a beneficial allele, based on present-day allele frequency patterns.

As ancient genome sequences become more readily available, they are increasingly being used to understand the process of natural selection (see reviews in **Malaspinas et al., 2012**; **Dehasque et al., 2020**). However, few studies have used ancient genomes to fit spatial dynamic models of the spread of an allele over a landscape. Most spatiotemporal analyses that included ancient genomes have used descriptive modeling in order to learn the spatiotemporal covariance structure of allele frequencies (**Segurel et al., 2020**) or hidden ancestry clusters (**Racimo et al., 2020b**), and then used that structure to hindcast these patterns onto a continuous temporally evolving landscape. In contrast to descriptive approaches, dynamic models have the power to infer interpretable parameters from genomic data and perhaps reveal the ultimate causes for these patterns (**Wikle et al., 2019**).

Dynamic models can also contribute to ongoing debates about the past trajectories of phenotypically important loci. For example, the geographic origin of the rs4988235(T) allele—upstream of the *LCT* gene and associated with adult lactase persistence in most of Western Eurasia (**Enattah et al., 2002**)—remains elusive, as is the way in which it spread (an extensive review can be found in **Ségurel and Bon, 2017**). The allele has been found in different populations, with frequencies ranging from 5% up to almost 100%, and its selection coefficient has been estimated to be among the highest in human populations (**Bersaglieri et al., 2004**; **Enattah et al., 2008**; **Tishkoff et al., 2007**). However, the exact causes for its adaptive advantage are contested (**Szpak et al., 2019**), and it has been suggested that

the selection pressures acting on the allele may have been different in different parts of the continent (*Gerbault et al., 2009*). Ancient DNA evidence shows that the allele was rare in Europe during the Neolithic (*Burger et al., 2007*; *Gamba et al., 2014*; *Allentoft et al., 2015*; *Mathieson et al., 2015*) and only became common in Northern Europe after the Iron Age, suggesting a rise in frequency during this period, perhaps mediated by gene flow from regions east of the Baltic where this allele was more common during the onset of the Bronze Age (*Krüttli et al., 2014*; *Margaryan et al., 2020*). *Itan et al., 2009* deployed their ABC approach to model the spatial spread of the rs4988235(T) allele and estimated that it was first under selection among farmers around 7500 years ago possibly between the Central Balkans and Central Europe. Others have postulated a steppe origin for the allele (*Allentoft et al., 2015*), given that the rise in frequency appears to have occurred during and after the Bronze Age migration of steppe peoples into Western Eurasia (*Haak et al., 2015*; *Allentoft et al., 2015*). However, the allele is at low frequency in genomes of Bronze Age individuals associated with Corded Ware and Bell Beaker assemblages in Central Europe who have high steppe ancestry (*Mathieson et al., 2015*; *Margaryan et al., 2020*), complicating the story further (*Ségurel and Bon, 2017*).

The origins and spread dynamics of large-effect pigmentation-associated SNPs in ancient Eurasians have also been intensely studied (*Ju and Mathieson, 2020*). Major loci of large effect on skin, eye, and hair pigmentation have been documented as having been under recent positive selection in Western Eurasian history (*Voight et al., 2006*; *Sabeti et al., 2007*; *Pickrell et al., 2009*; *Lao et al., 2007*; *Mathieson et al., 2015*; *Alonso et al., 2008*; *Hudjashov et al., 2013*). These include genes *SLC45A2*, *OCA2*, *HERC2*, *SLC24A5,* and *TYR*. While there is extensive evidence supporting the adaptive significance of these alleles, debates around their exact origins and spread are largely driven by comparisons of allele frequency estimates in population groups, which are almost always discretized in time and/or space. Among these, selection at the *TYR* locus is thought to have occurred particularly recently, over the last 5000 years (*Stern et al., 2019*), driven by a recent mutation (*Albers and McVean, 2020*) that may have spread rapidly in Western Eurasia.

Here, we develop a method to model the spread of a recently selected allele across both space and time, avoiding artificial discretization schemes to more rigorously assess the evidence for or against a particular dispersal process. We begin with the model proposed by *Novembre et al., 2005* and adapt it in order to handle ancient low-coverage genomic data and explore more complex models that allow for both diffusion and advection (i.e., directional transport) in the distribution of allele frequencies over space, as well as for a change in these parameters at different periods of time. We apply the method to alleles in two of the aforementioned loci in the human genome, which have been reported to have strong evidence for recent positive selection: *LCT/MCM6* and *TYR*. We focus on Western Eurasia during the Holocene, where ancient genomes are most densely sampled, and infer parameters relevant to the spread of these alleles, including selection, diffusion and advection coefficients.

## Results
### Summary of model
We based our statistical inference framework on a model proposed by *Novembre et al., 2005* to fit allele frequencies in two dimensions to present-day genotype data spread over a densely sampled map. We extend this model in several ways:

- We incorporate temporally sampled data (ancient genomes) to better resolve changes in frequency distributions over time.
- We make use of genotype likelihoods and pseudohaploid genotypes to incorporate low-coverage data into the inference framework.
- We permit more general dynamics by including advection parameters.
- We allow the selection, advection, and diffusion parameters to be different in different periods of time. Specifically, to reflect changes in population dynamics and mobility before and after the Bronze Age (*Loog et al., 2017*; *Racimo et al., 2020a*), we partitioned the model fit into two time periods: before and after 5000 years BP.

We explored the performance of two different spread models, which are extensions of the original model by *Novembre et al., 2005*, hereby called model A. This is a diffusion model containing a selection coefficient $s$ (determining the rate of local allele frequency growth) and a single diffusion term ($\sigma$). A more general diffusion model—hereby model B—allows for two distinct diffusion parameters

for latitudinal ($\sigma_y$) and longitudinal ($\sigma_x$) spread. Finally, model C is even more general and includes two advection terms ($v_x$ and $v_y$), allowing the center of mass of the allele's frequency to diverge from its origin over time. The incorporation of advection is meant to account for the fact that population displacements and expansions could have led to allele frequency dynamics that are poorly explained by diffusion alone.

In order to establish a starting time point for our diffusion process, we used previously published allele age estimates obtained from a nonparametric approach leveraging the patterns of haplotype concordance and discordance around the mutation of interest (*Albers and McVean, 2020*). In the case of the allele in the *LCT/MCM6* region, we also used age estimates based on an approximate Bayesian computation approach (*Itan et al., 2009*).

## Performance on deterministic simulations

To characterize the accuracy of our inference method under different parameter choices, we first generated deterministic simulations from several types of diffusion models. First, we produced an allele frequency surface map with a specified set of parameters from which we drew 1040 samples matching the ages, locations, and genotype calling format (diploid vs. pseudohaploid) of the 1040 genomes that we analyze below when studying the rs1042602(A) allele.

We generated six different simulations with different diffusion coefficients and afterward ran our method assuming model B. The results (simulations B1–B6) are summarized in *Figure 1*, *Figure 1—figure supplements 1–5*, and *Appendix 2—table 1*. Overall, the model is more accurate at correctly inferring the parameters for the time period before 5000 years BP (*Figure 1b*), with decreased performance when longitudinal diffusion is high (*Figure 1—figure supplement 5*).

Next, we investigated the performance of model C, which includes advection coefficients. We generated four different simulations including advection (simulations C1–C4: *Figure 2*, *Figure 1—figure supplements 1–3*, and *Appendix 2—table 2*). We found that our method is generally able to estimate the selection coefficient accurately. However, in some of the simulations, we found discrepancies between the estimated and true diffusion and advection coefficients, often occurring because of a misestimated origin forcing the other parameters to adjust in order to better fit the allele frequency distribution in later stages of the allele's spread (*Figure 2*). Despite the disparities between the true and inferred parameter values, the resulting surface plots become very similar as we approach the present, suggesting that different combinations of parameters can produce similar present-day allele frequency distributions.

## Advection model applied to non-advection simulations

We assessed model performance when we apply model C, which includes advection coefficient estimates, to simulations generated without advection (see *Figure 1—figure supplements 6 and 7*). We can observe that the advection coefficients are inferred to be non-zero (*Figure 1—figure supplements 6b and 7b*); however, the inferred allele frequency dynamic plots closely resemble the ones obtained with true parameter values (*Figure 1—figure supplements 6a and 7a*). This shows that complex interactions between the diffusion and advection coefficients can result in similar outcomes even when only diffusion is considered in the model.

The inference of the origin of the allele also differs when we compare results when using models B and C. In order to understand better how the model estimates the allele origin, we highlighted the first individual in simulations B1 and B4 that contains the derived allele. We can see that in the case of simulation B1 the inferred origin of the allele is close to the first observance of the derived allele in the model that includes advection. In contrast when the advection is not included, the origin of the allele is inferred to be closer to where it is initially rising in frequency (*Figure 1—figure supplements 1a and 4a*). However, this is not always the case. For instance, if we look at the results from the advection model on simulation B4, we can see that the origin of the allele is inferred relatively far from the sample known to have carried the first instance of the derived allele. Therefore, if there is a relatively large interval between the time when the allele originated and when the first ancient genomes are available, the beneficial allele can spread widely, but as this spread is not captured by any of the data points, inference of the precise origin of the selected allele is nearly impossible.

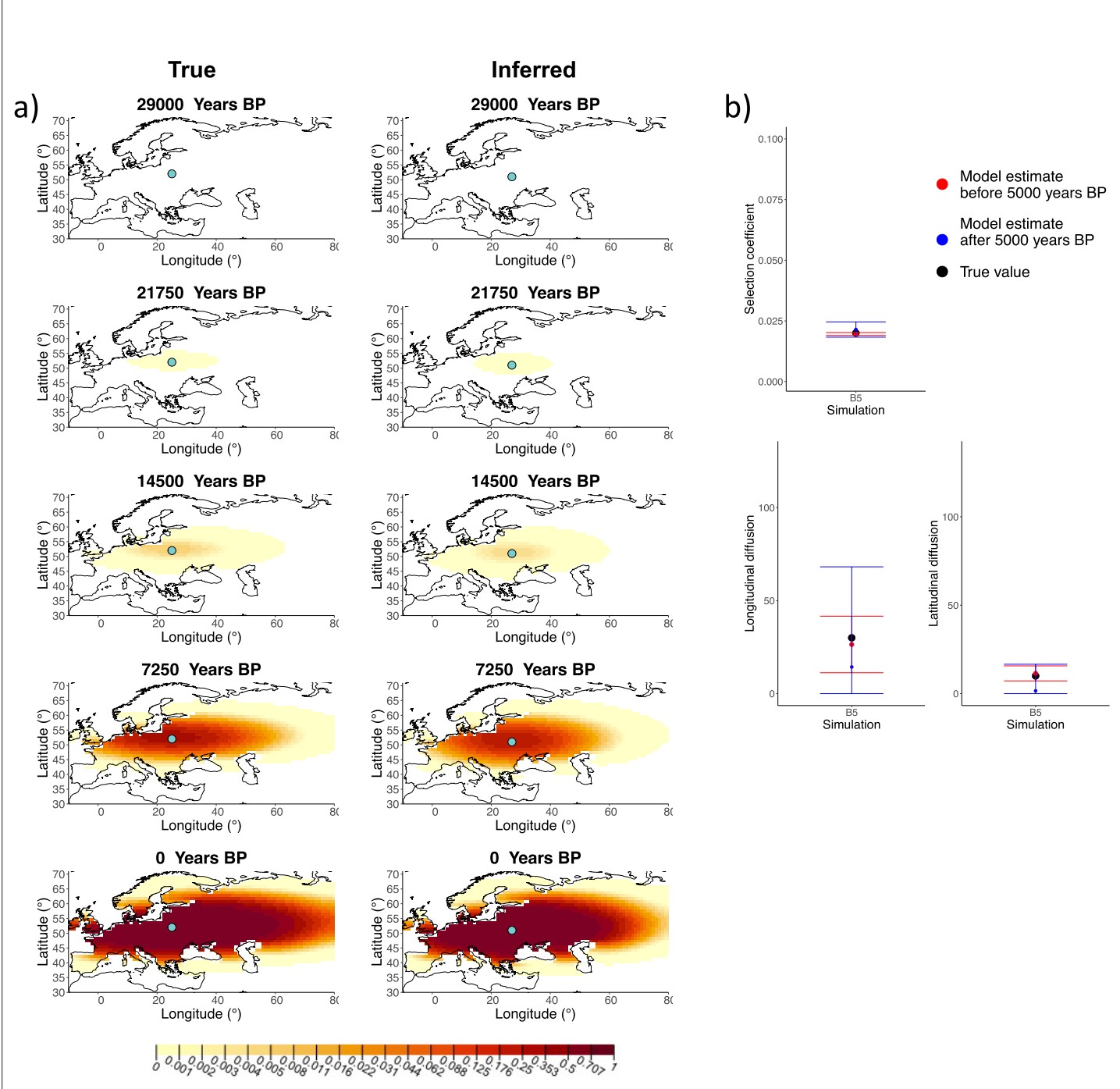

**Figure 1.** Comparison of true and inferred allele frequency dynamics for simulation B5. (**a**) Comparison of true and inferred allele frequency dynamics for a simulation with diffusion and no advection (B5). The green dot corresponds to the origin of the allele. The parameter values used to generate the frequency surface maps are summarized in *Appendix 2—table 1*. (**b**) Comparison of true parameter values and model estimates. Whiskers represent 95% confidence intervals.

The online version of this article includes the following figure supplement(s) for figure 1:

**Figure supplement 1.** Comparison of true and inferred allele frequency dynamics for simulation B1.

**Figure supplement 2.** Comparison of true and inferred allele frequency dynamics for simulation B2.

**Figure supplement 3.** Comparison of true and inferred allele frequency dynamics for simulation B3.

**Figure supplement 4.** Comparison of true and inferred allele frequency dynamics for simulation B4.

*Figure 1 continued on next page*

*Figure 1 continued*

**Figure supplement 5.** Comparison of true and inferred allele frequency dynamics for simulation B6.

**Figure supplement 6.** Comparison of true allele frequency dynamics for simulation B1 and those inferred by the model C.

**Figure supplement 7.** Comparison of true allele frequency dynamics for simulation B4 and those inferred by the model C.

## Impact of sample clustering on parameter estimates

We evaluated the impact of different sampling and clustering schemes on our inferences that could potentially arise by aggregating aDNA data from studies with different sampling schemes. We used a deterministic simulation to create three different degrees of clustering, which we will refer to as 'homogeneous,' 'intermediate,' or 'extreme' by varying the area from which we sample individuals to be used in our inferences (*Figure 3—figure supplement 1*). Additionally, we also tested the impact of biased temporal sampling in the periods before and after 5000 years BP by oversampling in the ancient period (75%/25%), equally sampling in the two periods (50%/50%), and oversampling in the recent period (25%/75%). Because we evaluated this temporal bias for each of the three spatial clustering sampling scenarios, this resulted in a total of nine different sampling scenarios. We note that the third 'extreme' spatial clustering scenario is completely unrealistic and one would not expect inferences of any degree of accuracy from it, but we believe it gives a good idea of the behavior of our method in the limiting case of extremely restricted spatial sampling.

A comparison of allele frequency maps generated using true parameter values and using parameter estimates from the different sampling schemes is shown in *Figure 3—figure supplements 2–9*. In *Figure 3* we show the allele frequency map generated using the 'intermediate 75%/25%' clustering scheme. Parameter estimates used to generate all these figures are summarized in *Appendix 2—table 3*. Overall we can see that the allele frequency maps inferred from these scenarios closely resemble the maps generated using the true parameter values, despite the challenges in finding accurate values for the individual point estimates of some of the parameters, highlighting that various combinations of diffusion and advection coefficients can produce similar underlying frequency maps (as discussed in the section 'Performance on deterministic simulations'). This suggests that the joint spatiotemporal information encoded in the inferred maps (not just the individual parameters estimates) should be used in interpreting model outputs, particularly when it comes to the advection and diffusion parameters. The selection coefficient estimates are inferred highly accurately, regardless of the sampling scheme chosen, and lie close to the true value, with only a slight underestimation in the time period after 5000 years BP (with the exception of the 'extreme 25%/75%').

## Spatially explicit forward simulations

In addition to drawing simulated samples from a diffusion model, we used SLiM (*Haller and Messer, 2019*) to perform spatially explicit individual-based forward-in-time simulations of selection acting on a beneficial allele by leveraging an R interface for spatial population genetics now implemented in the R package *slendr* (*Petr, 2021*).

We introduced a single beneficial additive mutation in a single individual and let it evolve across the European landscape. Before applying our method on the simulated data, we sampled 1040 individuals whose ages were log-uniformly distributed to ensure that there were more samples closer to the present, as in the real data. We transformed the diploid genotypes to pseudohaploid genotypes by assigning a heterozygous individual an equal probability of carrying the ancestral or the derived genotype. The parameter values estimated by our model to the simulations described in this section are summarized in *Appendix 2—table 4*.

We can see that the origin of the allele inferred by the model closely corresponds to the first observation of the derived allele in the simulation (*Figure 4*). The inferred selection coefficient is only slightly higher than the true value from the simulation (0.0366 vs. 0.030). In general, the model accurately captures the spread of the allele centered in Central Europe, though we observe some discrepancies due to differences between the model assumed in the simulation (which, e.g., accounts for local clustering of individuals, *Figure 4—figure supplement 1*), and that assumed by our diffusion-based inference.

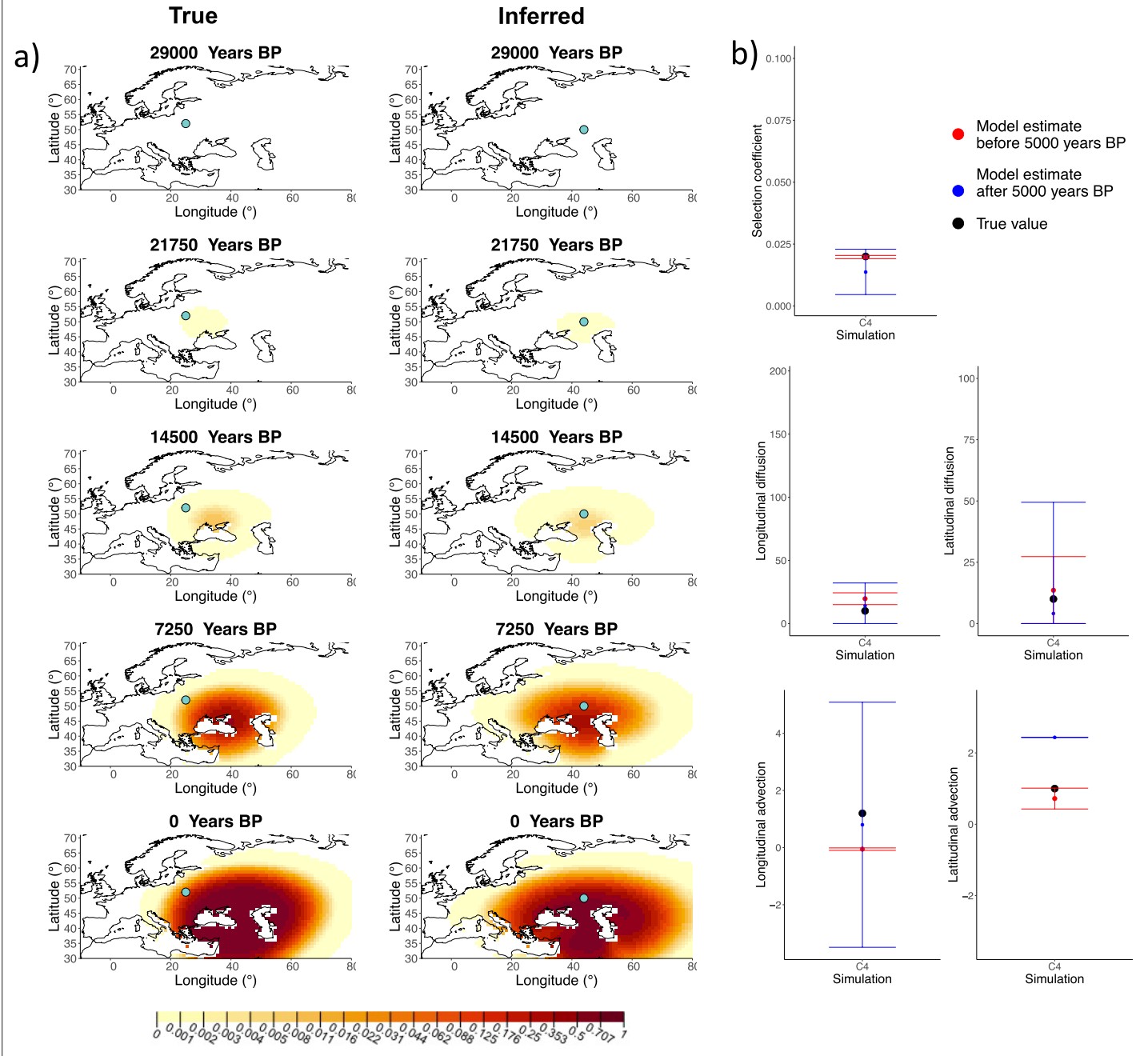

**Figure 2.** Comparison of true and inferred allele frequency dynamics for simulation C4. (**a**) Comparison of true and inferred allele frequency dynamics for one of the simulations including advection (C4). The green dot corresponds to the origin of the allele. The parameter values used to generate the frequency surface maps are summarized in *Appendix 2—table 2*. (**b**) Comparison of true parameter values and model estimates. Whiskers represent 95% confidence intervals.

The online version of this article includes the following figure supplement(s) for figure 2:

**Figure supplement 1.** Comparison of true and inferred allele frequency dynamics for simulation C1.

**Figure supplement 2.** Comparison of true and inferred allele frequency dynamics for simulation C2.

**Figure supplement 3.** Comparison of true and inferred allele frequency dynamics for simulation C3.

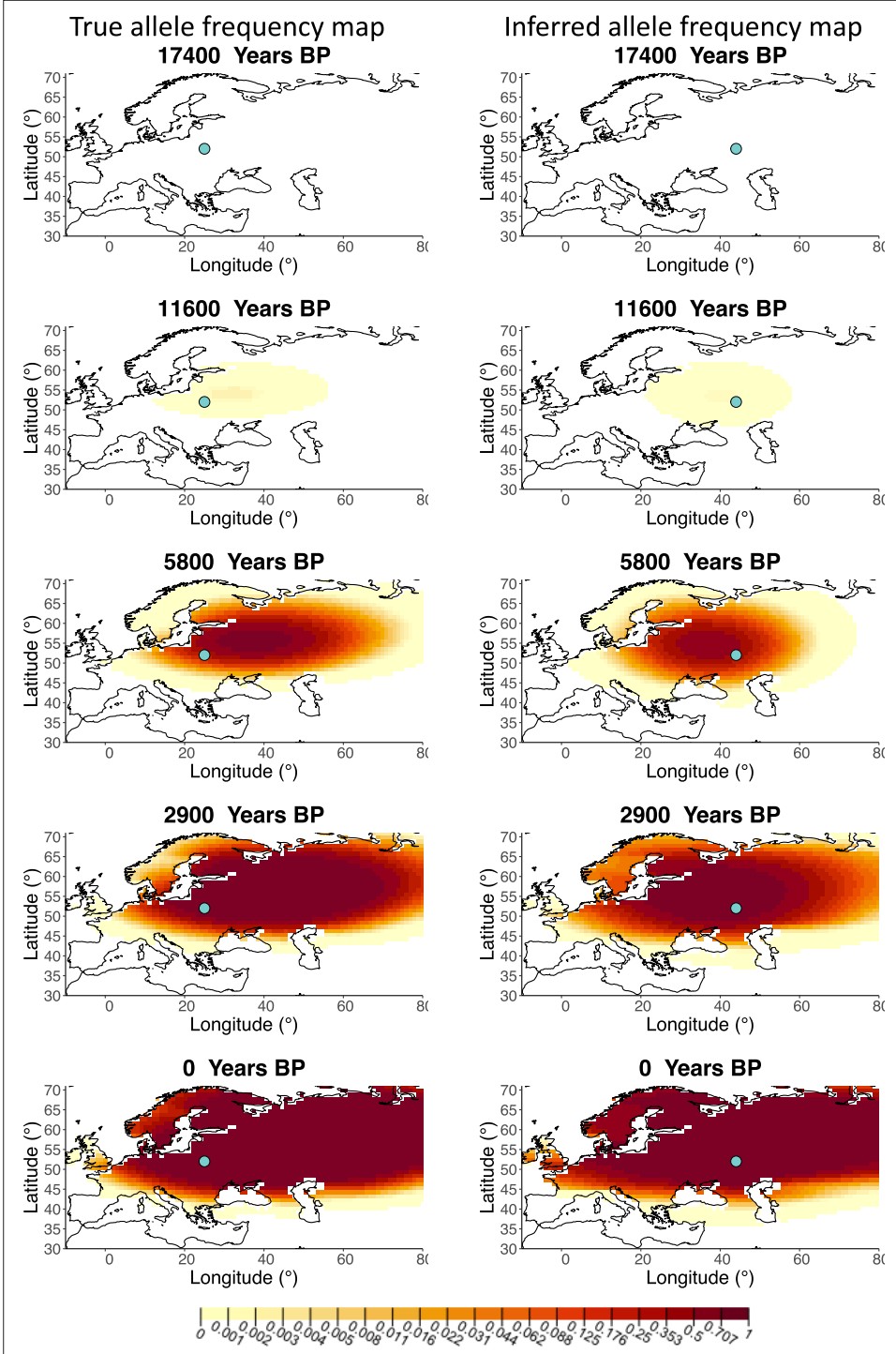

**Figure 3.** Comparison of true allele frequency map and map generated using 'intermediate 75%/25%' clustering scheme. Left: allele frequency map generated using true parameter values. Right: allele frequency map generated using parameter estimates for 'intermediate 75%/25%' clustering scheme. Parameter values used to generate the maps are summarized in *Appendix 2—table 3*.

The online version of this article includes the following figure supplement(s) for figure 3:

**Figure supplement 1.** Examples of spatial sampling scenarios for each of the three clustering schemes.

**Figure supplement 2.** Allele frequency map generated using true parameter values and using parameter estimates for 'homogeneous 75%/25%' clustering scheme.

*Figure 3 continued on next page*

*Figure 3 continued*

**Figure supplement 3.** Allele frequency map generated using true parameter values and using parameter estimates for 'homogeneous 50%/50%' clustering scheme.

**Figure supplement 4.** Allele frequency map generated using true parameter values and using parameter estimates for 'homogeneous 25%/75%' clustering scheme.

**Figure supplement 5.** Allele frequency map generated using true parameter values and using parameter estimates for 'intermediate 50%/50%' clustering scheme.

**Figure supplement 6.** Allele frequency map generated using true parameter values and using parameter estimates for 'intermediate 25%/75%' clustering scheme.

**Figure supplement 7.** Allele frequency map generated using true parameter values and using parameter estimates for 'extreme 75%/25%' clustering scheme.

**Figure supplement 8.** Allele frequency map generated using true parameter values and using parameter estimates for 'extreme 50%/50%' clustering scheme.

**Figure supplement 9.** Allele frequency map generated using true parameter values and using parameter estimates for 'extreme 25%/75%' clustering scheme.

## Dynamics of the rs4988235(T) allele

Having tested the performance of our method on simulated data, we set out to infer the allele frequency dynamics of the rs4988235(T) allele (associated with adult lactase persistence) in ancient Western Eurasia. For our analysis, we used a genotype dataset compiled by *Segurel et al., 2020*, which amounts to 1434 genotypes from ancient Eurasian genomes individuals, and a set of 36,659 genotypes from present-day Western and Central Eurasian genomes (Ségurel and Bon, 2017; *Heyer et al., 2011*; *Marchi et al., 2018*; *Liebert et al., 2017*; *Gallego Romero et al., 2012*; *Itan et al., 2010*; *Charati et al., 2019*). After filtering out individuals falling outside of the range of the geographic boundaries considered in this study, we retained 1332 ancient individuals. The locations of ancient and present-day individuals used in the analysis to trace the spread of rs4988235(T) are shown in *Figure 5*.

We used a two-period scheme by allowing the model to have two sets of estimates for the selection coefficient and the diffusion and advection coefficients in two different periods of time: before and after 5000 years ago, reflecting the change in population dynamics and mobility before and after the Bronze Age transition (*Loog et al., 2017*; *Racimo et al., 2020a*). We used two allele age estimates as input: a relatively young one (7441 years ago) obtained by using the estimated start of selection onset from *Itan et al., 2009* (though we note this is necessarily a lower bound of the age of mutation origin), and a relatively old one (20,106 years ago) obtained from the age estimate from *Albers and McVean, 2020*. The results obtained for fitting the model on rs4988235(T) are summarized in *Appendix 2—table 5* and *Appendix 2—table 6*, and in *Figure 6b* (younger age) and *Figure 6— figure supplement 1* (older age).

Assuming the mutation age estimate is equivalent to the start of selection onset from *Itan et al., 2009*, the origin of the allele is estimated to be north of the Caucasus, around what is now south-western Russia and eastern Ukraine (*Figure 6b*). Given that this age is relatively young, our method fits a very strong selection coefficient ($\approx 0.1$) during the first period in order to accommodate the early presence of the allele in various points throughout Eastern Europe, and a weaker (but still strong) selection coefficient ($\approx 0.03$) in the second period. We also estimate stronger diffusion in the second period than in the first, to accommodate the rapid expansion of the allele throughout Western Europe, and a net westward advection parameter, indicating movement of the allele frequency's center of mass to the west as we approach the present.

Assuming the older age estimate from *Albers and McVean, 2020*, the origin of the allele is estimated to be in the northeast of Europe (*Figure 6—figure supplement 1*), which is at a much higher latitude than the first occurrence of the allele, in Ukraine. Due to the deterministic nature of the model, the frequency is implicitly imposed to expand in a region where there are no actual observed instances of the allele. The model compensates for this by placing the origin in an area with a lower density of available aDNA data and thus avoiding an overlap of the increasing allele frequencies with individuals who do not carry the derived rs4988235(T) allele (see *Figure 6a*). As the model expands rapidly in the southern direction (*Appendix 2—table 6*) it eventually reaches the sample carrying the derived variant in Ukraine.

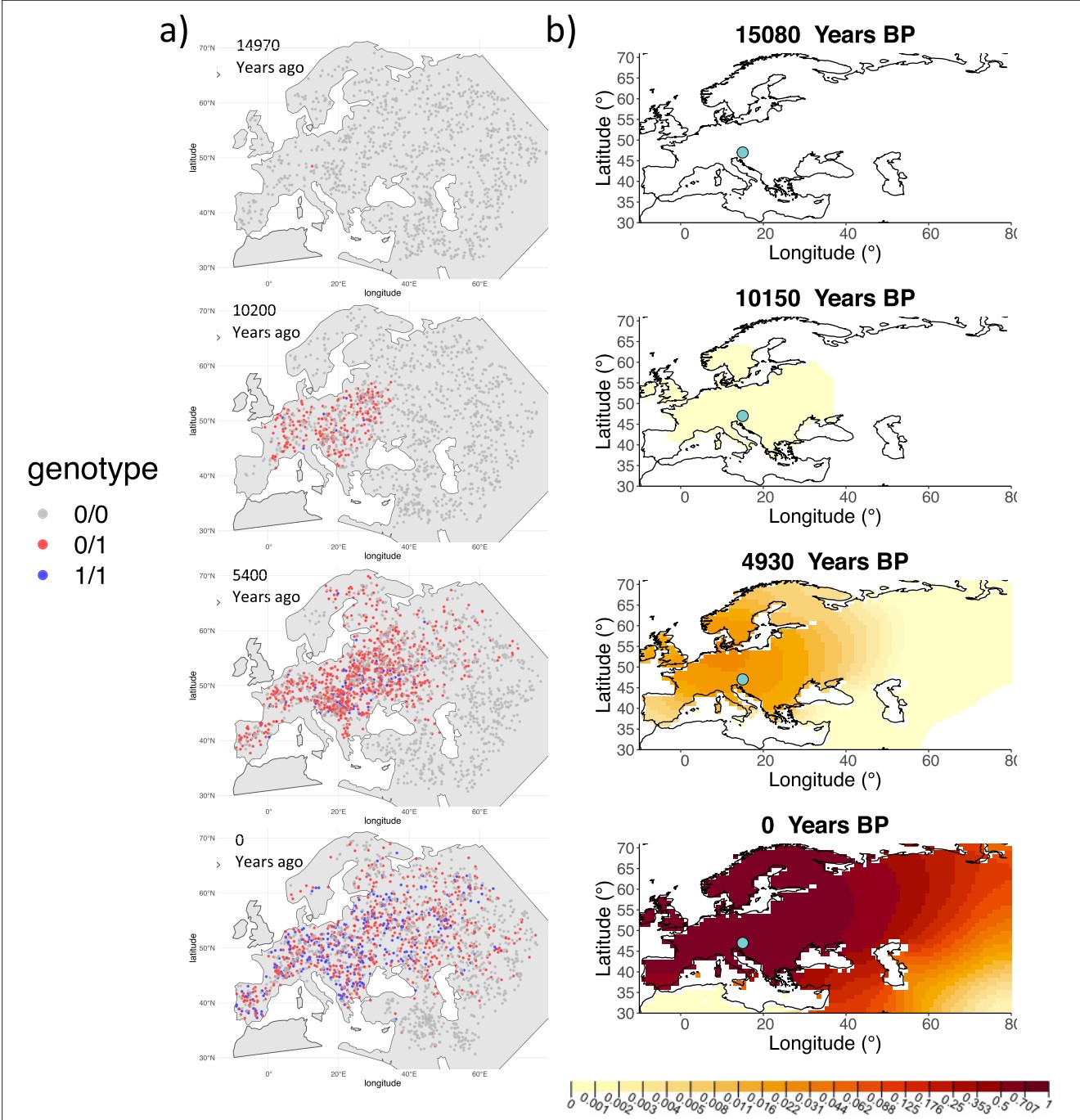

**Figure 4.** Comparison of an individual-based simulation and allele frequency dynamics inferred by the diffusion model. (**A**) Individual-based simulation of an allele that arose in Central Europe 15,000 years ago with a selection coefficient of 0.03. Each dot represents a genotype from a simulated genome. To avoid overplotting, only 1000 out of the total 20,000 individuals in the simulation in each time point are shown for each genotype category. (**B**) Allele frequency dynamics inferred by the diffusion model on the individual-based simulation to the left, after randomly sampling 1040 individuals from the simulation and performing pseudohaploid genotype sampling on them. The ages of sampled individuals were log-uniformly distributed. The estimated parameter values of the fitted model are shown in *Appendix 2—table 4*.

The online version of this article includes the following figure supplement(s) for figure 4:

**Figure supplement 1.** Distribution of individuals across the map under neutrality, showing the tendency of individuals to cluster together.

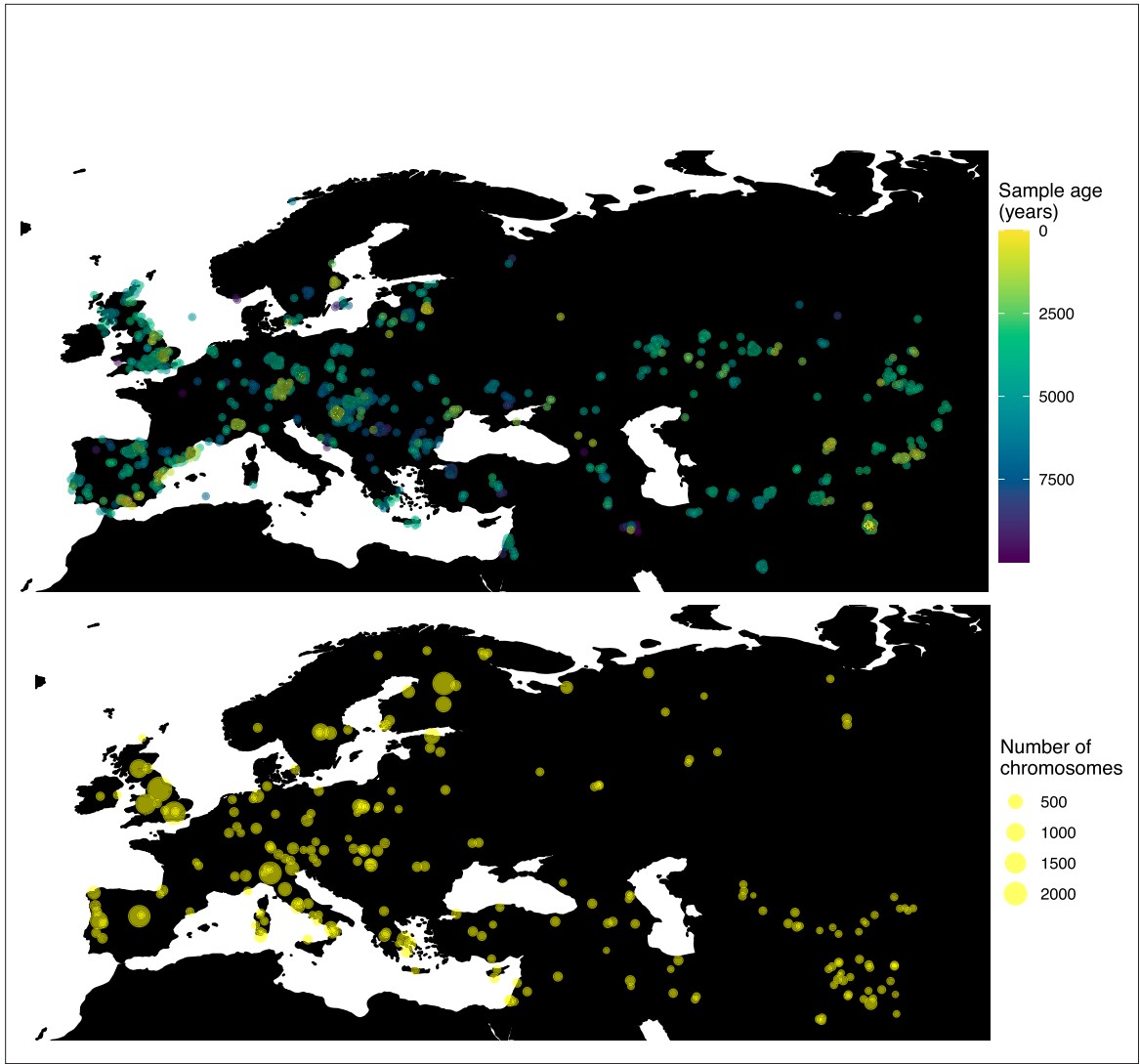

**Figure 5.** Locations of samples used to model the spread of the rs4988235(T) allele. The upper panel shows the spatiotemporal locations of ancient individuals, and the bottom panel represents the locations of present-day individuals.

## Dynamics of the rs1042602(A) allele

Next, we investigated the spatiotemporal dynamics of the spread of an allele at a pigmentation-associated SNP in the *TYR* locus (rs1042602(A)), which has been reported to be under recent selection in Western Eurasian history (*Stern et al., 2019*). For this purpose, we applied our method to the Allen Ancient DNA Resource data (*Reich and Mallick, 2019*), which contains randomly sampled pseudohaploid genotypes from 1513 published ancient Eurasian genomes (listed in *Supplementary file 1*), from which we extracted those that had genotype information at this locus in Western Eurasia. We merged this dataset with diploid genotype information from high-coverage present-day West Eurasian genomes from the Human Genome Diversity Panel (HGDP) (*Bergström et al., 2020*), which resulted in a total of 1040 individuals with genotype information at rs1042602, which were used as input to our analysis. Geographic locations of individuals in the final dataset are shown in *Figure 7*.

Similarly to our analysis of the spread of the allele in rs4988235(T), we inferred the dynamics of the rs1042602(A) allele separately for the time periods before and after 5000 years BP and assuming the age of the allele to be 26,361 years (*Albers and McVean, 2020*). The inferred parameters for both time periods are summarized in *Appendix 2—table 7*, and the allele frequency surface maps generated using these parameters are shown in *Figure 8b*. The origin of the rs1042602(A) corresponds closely to the region where the allele initially starts to segregate in the time period between 7500

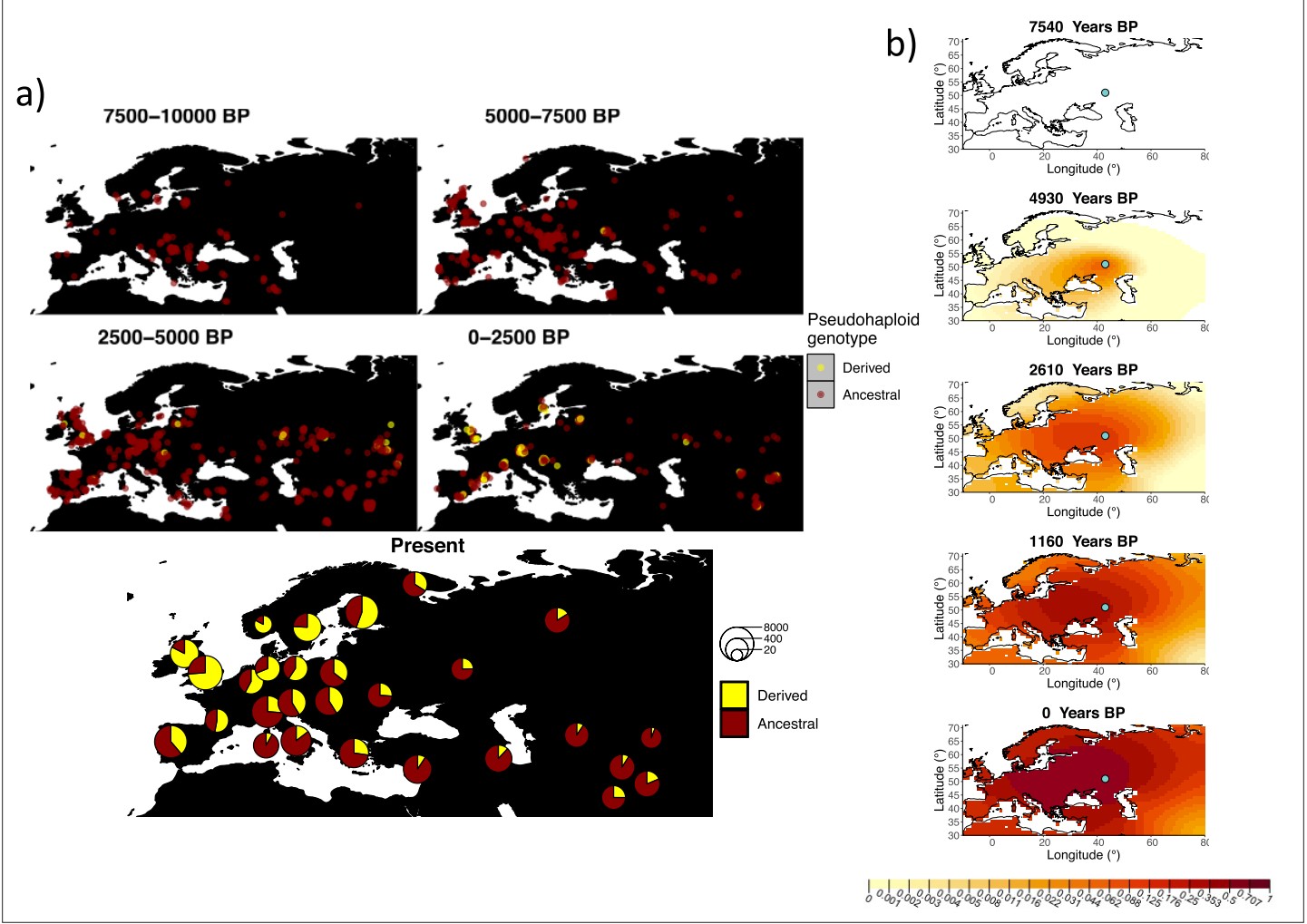

**Figure 6.** Allele frequency dynamics of rs4988235(T). (**a**) Top: pseudohaploid genotypes of ancient samples at the rs4988235 SNP in different periods. Yellow corresponds to the rs4988235(T) allele. Bottom: allele frequencies of present-day samples represented as pie charts. The size of the pie charts corresponds to the number of available sequences in each region. (**b**) Inferred allele frequency dynamics of rs4988235(T). The green dot indicates the inferred geographic origin of the allele.

The online version of this article includes the following figure supplement(s) for figure 6:

**Figure supplement 1.** Inferred frequency dynamics of rs4988235(T) using the allele age that was inferred in *Albers and McVean, 2020*.

**Figure supplement 2.** Inferred frequency dynamics of rs4988235(T) when the origin of the allele is moved 10° west from the original estimate.

**Figure supplement 3.** Inferred frequency dynamics of rs4988235(T) when the origin of the allele is moved 10° east from the original estimate.

**Figure supplement 4.** Inferred frequency dynamics of rs4988235(T) when the origin of the allele is moved 10° north from the original estimate.

**Figure supplement 5.** Inferred frequency dynamics of rs4988235(T) when the origin of the allele is moved 10° south from the original estimate.

**Figure supplement 6.** Inferred frequency dynamics of rs4988235(T) forcing the geographic origin of the allele to be at the location inferred in *Itan et al., 2009*.

**Figure supplement 7.** Inferred frequency dynamics of rs4988235(T) assuming the allele age to be the lower end of the 95% credible interval for the start of selection onset inferred in *Itan et al., 2009*.

**Figure supplement 8.** Inferred frequency dynamics of rs4988235(T) assuming the allele age to be the higher end of the 95% credible interval for the start of selection onset inferred in *Itan et al., 2009*.

**Figure supplement 9.** Log-likelihood values for model runs using different ages of the rs4988235(T) allele as input, with the age inferred by *Itan et al., 2009* we use as fixed input highlighted in red.

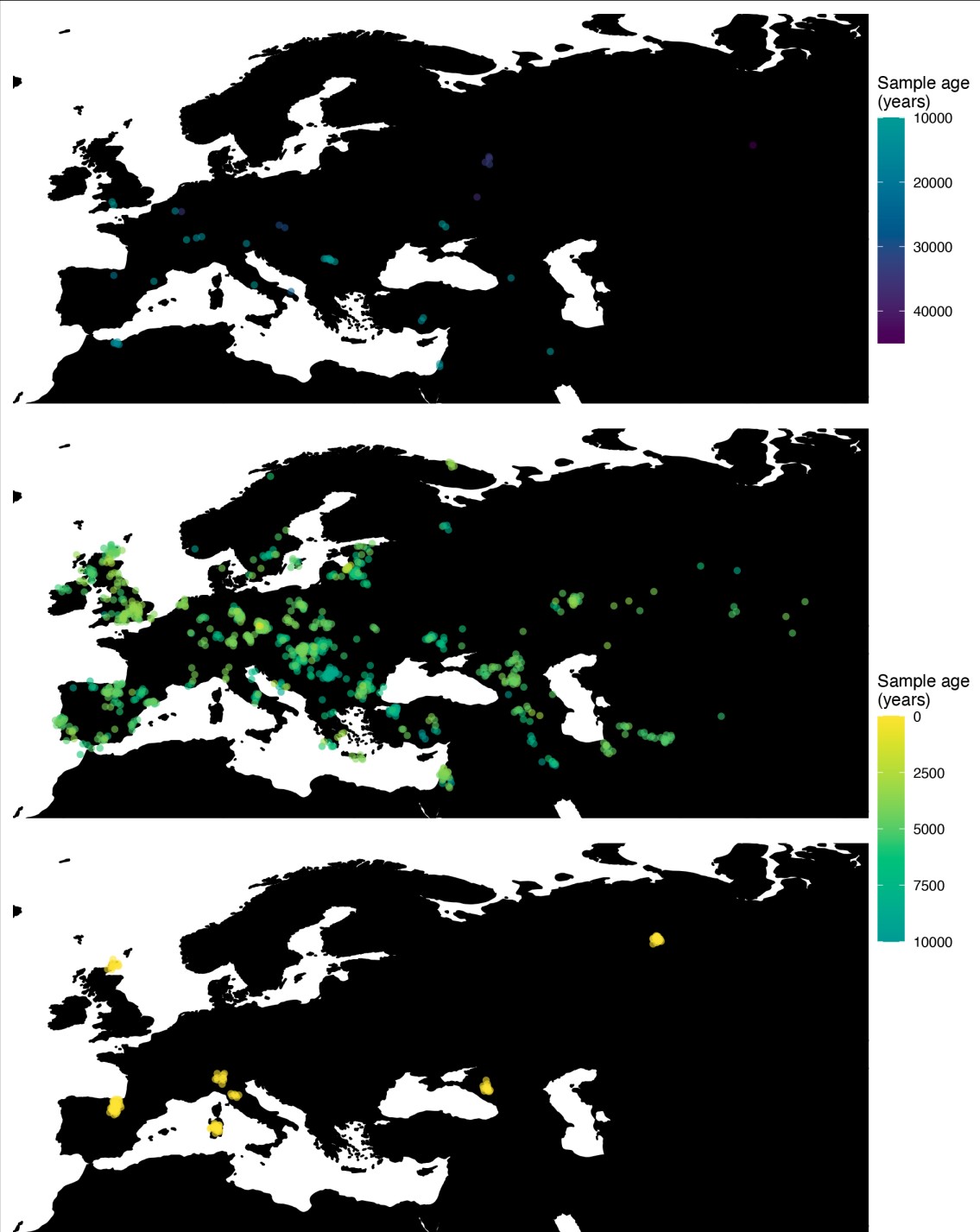

**Figure 7.** Spatiotemporal sampling locations of sequences used to model the rs1042602(A) allele in Western Eurasia. Upper panel: ancient individuals dated as older than 10,000 years ago. Middle panel: ancient individuals dated as younger than 10,000 years ago. Bottom panel: present-day individuals from the Human Genome Diversity Panel (HGDP).

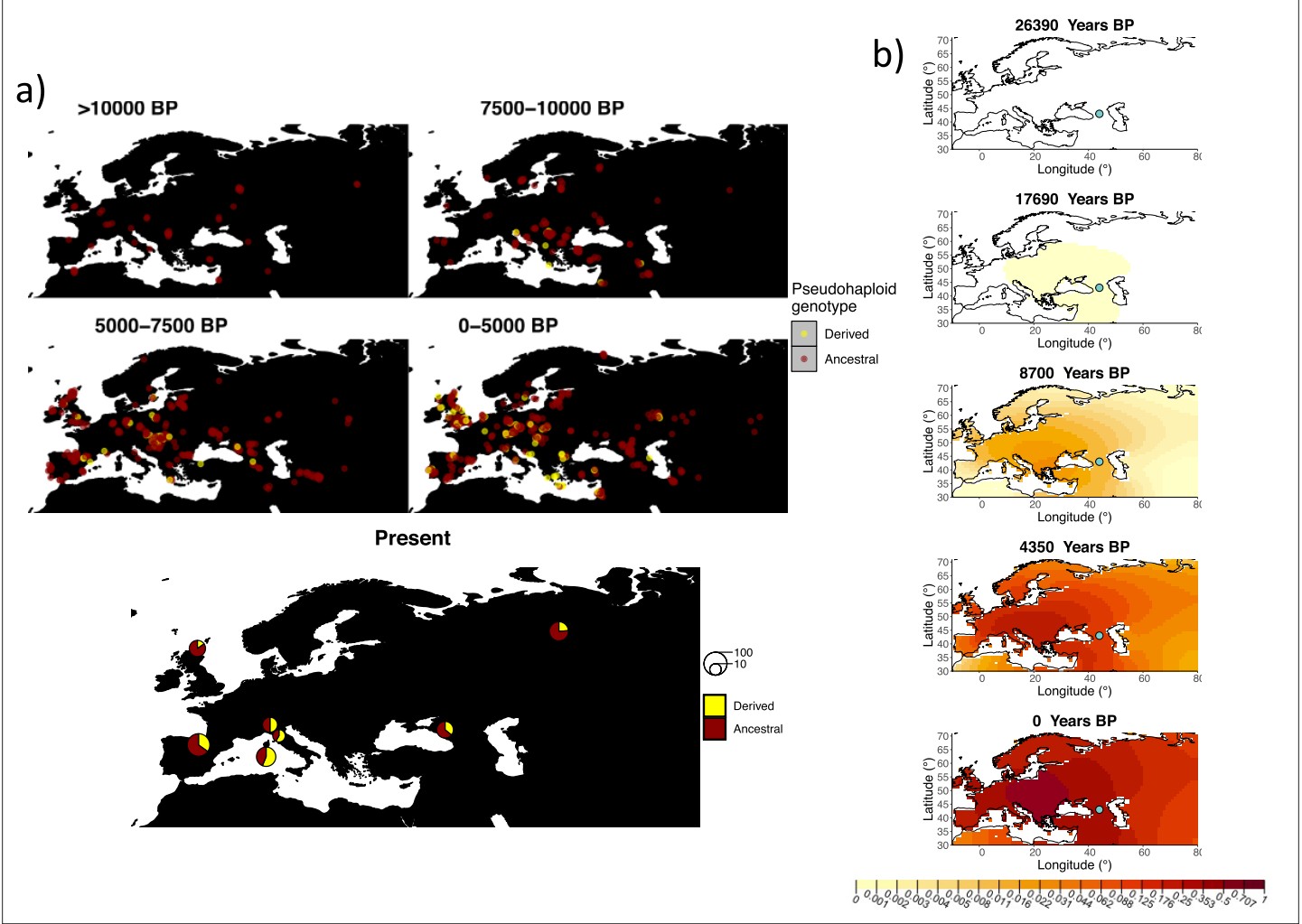

**Figure 8.** Allele frequency dynamics of rs1042602(A). (**a**) Top: pseudohaploid genotypes of ancient samples of the rs1042602 in different periods. Yellow corresponds to the A allele. Bottom: diploid genotypes of present-day samples. (**b**) Inferred allele frequency dynamics of rs1042602(A). The green dot corresponds to the inferred geographic origin of the allele.

The online version of this article includes the following figure supplement(s) for figure 8:

**Figure supplement 1.** Inferred frequency dynamics of rs1042602(A) when the origin of the allele is moved 10° east from the original estimate.

**Figure supplement 2.** Inferred frequency dynamics of rs1042602(A) when the origin of the allele is moved 10° north from the original estimate.

**Figure supplement 3.** Inferred frequency dynamics of rs1042602(A) when the origin of the allele is moved 10° south from the original estimate.

**Figure supplement 4.** Inferred frequency dynamics of rs1042602(A) assuming the allele age to be the lower end of the 95% confidence interval for the allele age inferred in *Albers and McVean, 2020*.

**Figure supplement 5.** Frequency dynamics of rs1042602(A) assuming the allele age to be the higher end of the 95% confidence interval for the allele age inferred in *Albers and McVean, 2020*.

**Figure supplement 6.** Log-likelihood values for model runs using different ages of the rs1042602(A) allele as input, with the age inferred by *Albers and McVean, 2020* we use as fixed input highlighted in red.

and 10,000 years BP as seen in *Figure 8a*. Estimates of the selection coefficient for both time periods (0.0221 and 0.0102 for the period before and after 5000 years BP, respectively) suggest that selection acting on the allele has decreased after 5000 years BP.

## Robustness of parameters to the inferred geographic origin of allele

We carried out an analysis to characterize how sensitive the selection, diffusion, and advection parameters are to changes in the assumed geographic origin of the allele. For the rs4988235(T) allele,

we forced the origin of the allele to be 10° away from our inferred origin in each cardinal direction, while assuming the allele age is equal to the inferred start of selection onset from *Itan et al., 2009* (*Appendix 2—table 8*). In *Figure 6—figure supplements 2–5*, we can see the allele frequency dynamics of these four scenarios, respectively. We also forced the allele origin to be at the geographic origin estimated in *Itan et al., 2009* (*Figure 6—figure supplement 6*, *Appendix 2—table 9*), which is westward of our estimate. In all five cases during the period prior 5000 years BP, the allele is inferred to expand in the direction of the first sample that is observed to carry the rs4988235(T) allele and is located in Ukraine. During the time period after 5000 years BP, the patterns produced by the model are rather similar, although the parameters associated with diffusion and advection differ, in order to account for the different starting conditions.

We also investigated how the results are affected when the estimated geographic origin of the rs1042602(A) allele is moved with respect to the initial estimate. We set the allele to be 10° east, 10° north, and 10° south of the original estimate as shown in *Figure 8—figure supplements 1–3*, respectively (for parameter estimates see *Appendix 2—table 10*). We did not look at a scenario in which the origin of the allele is moved to the west since it would either end up in the Black Sea or more westward than 10°. The selection coefficient remains similar to the original estimate throughout all three scenarios. The way the allele spreads across the landscape is also similar in all cases and, as in the case of rs4988235(T), the model accounts for the different origins of the allele by adjusting the diffusion and advection coefficients in the time period after 5000 years BP.

### Robustness of parameters to the assumed age of the allele

In order to investigate how sensitive our inferences are to the point estimates of allele ages we obtained from the literature (*Albers and McVean, 2020*; *Itan et al., 2009*), we also fitted our model using the upper and lower ends of the 95% confidence intervals or credible intervals for each age estimate (depending on whether the inference procedure in the literature was via a maximum likelihood or a Bayesian approach). For the rs4988235(T) allele, the reported credible intervals for the (*Itan et al., 2009*) age are 8683 and 6256 years BP. For the rs1042602(A) allele, the reported confidence intervals for the age are 27,315 and 25,424 years BP (*Albers and McVean, 2020*).

When refitting the model for the rs4988235(T) allele, we found that the inferred selection coefficient is slightly lower when the allele age is assumed to be at the lower bound of the 95% credible interval (0.0867 vs. 0.0993 before 5000 years BP and 0.0321 vs. 0.0328 after 5000 years BP) and slightly higher when assumed to be at the upper bound (0.0994 vs. 0.0993 before 5000 years BP and 0.0572 vs. 0.0328 after 5000 years BP) (*Appendix 2—table 5* and *Figure 6—figure supplements 7 and 8*). This occurs because the selection intensity must be higher or lower when there is more or less time, respectively, for the allele to reach the allele frequencies observed in the data. In the case of the rs1042602(A) allele, this only affects the earlier time period (*Appendix 2—table 7*). The rs4988235(T) allele's geographic distribution in the more recent time periods is also less extended geographically when the age is assumed to be young. The inferred geographic origin of both alleles slightly differs under different assumed ages (*Figure 8—figure supplements 4 and 5*).

In addition, we assessed the likelihood of the best-fitted models with varying the ages of the rs4988235(T) and rs1042602(A) alleles (*Figure 6—figure supplement 9* and *Figure 8—figure supplement 6*, respectively). We can see that in the case of rs4988235(T) allele the allele age used in this study (7441 years) gives the most likely solution among the explored ages. In case of the rs1042602(A) allele, we found that there are multiple nearly equally likely ages when looking at ages at least as old as 15,000 years.

## Discussion

A spatially explicit framework for allele frequency diffusion can provide new insights into the dynamics of selected variants across a landscape. We have shown that under the conditions of strong, recent selection, our method can infer selection and dispersal parameters using a combination of ancient and present-day human genomic data. However, when allowing for advection, the inferred location tends to become less accurate. This suggests that migration events early in the dispersal of the selected allele could create difficulties in finding the true allele origin if net directional movement (i.e., via major migratory processes) had a large effect in this dispersal. This issue could be alleviated with the

inclusion of more ancient genomes around the time of the origin of the mutation, perhaps in combination with a more fine-scaled division into periods where advection may have occurred in different directions.

The inferred geographic origin of the rs4988235(T) allele reflects the best guess of our framework given the constraints provided by its input, namely, the previously inferred age of the allele and the observed instances of this allele throughout Western Eurasia. We are also assuming that the allele must have arisen somewhere within the bounding box of our studied map. When assuming a relatively young allele age (7441 years ago, equal to the start of selection onset in *Itan et al., 2009*), the origin of the allele is placed north of the Caucasus, perhaps among steppe populations that inhabited the area at this time (*Haak et al., 2015*; *Allentoft et al., 2015*). This origin is further east than the geographic origin estimate from *Itan et al., 2009*, likely reflecting additional ancient DNA information that is available to us, and indicates an early presence of the allele in Eastern Europe. When assuming a relatively old allele age (20,106 years ago, *Albers and McVean, 2020*), the age is placed in Northeast Europe, perhaps among Eastern hunter-gatherer groups that inhabited the region in the early Holocene. We note that the number of available genomes for Eastern and Northeastern Europe during the early Holocene is scarce, so the uncertainty of the exact location of this origin is relatively high. Regardless of the assumed age, we estimate a net westward displacement of the allele frequency's center of mass, and a rapid diffusion, particularly in the period after 5000 years ago.

Various studies have estimated the selection coefficient for the rs4988235(T) allele, and these range from as low as 0.014 to as high as 0.19 (*Enattah et al., 2008*; *Mathieson and Mathieson, 2018*; *Mathieson, 2020*; *Stern et al., 2019*; *Burger et al., 2020*; *Peter et al., 2012*; *Gerbault et al., 2009*; *Itan et al., 2009*; *Bersaglieri et al., 2004*). Recent papers incorporating ancient DNA estimate the selection coefficient to be as low as 0 (in certain regions of Southern Europe) and as high as 0.06 (*Mathieson and Mathieson, 2018*; *Mathieson, 2020*; *Burger et al., 2020*). It is also likely that the selection coefficient was different for different regions of Europe, perhaps due to varying cultural practices (*Mathieson, 2020*). In our case, the estimated selection coefficient during the first period—before 5000 years ago—depends strongly on the assumed allele age (s = 0.0993 vs. s = 0.0285). As in the case of the geographic origin, these estimates should be taken with caution as the number of available allele observations in the early Holocene is fairly low. The estimates for the second period—after 5000 years ago—are more robust to the assumed age: s = 0.0328 (95% CI: 0.0327–0.0329) if we assume the younger allele age (7441 years ago) and s = 0.0255 (95% CI: 0.0252–0.0258) if we assume the older allele age (20,106 years ago). These estimates are also within the range of previous estimates.

In the case of the rs1042602(A) allele, our estimated selection coefficients of 0.0221 (95% CI: 0.0216–0.0227) and 0.0102 (95% CI: 0.0083–0.0120) for the time periods before and after 5000 years BP, respectively, are generally in agreement with previous results. *Wilde et al., 2014* used a forward simulation approach to infer a point estimate of 0.026. Another study using an approximate Bayesian computation framework (*Nakagome et al., 2019*) estimated the strength of selection acting on rs1042602 to be 0.013 (0.002–0.029). Although both studies utilized ancient DNA data, the estimates were obtained without explicitly modeling the spatial dimension of the selection process.

Our estimates of the longitudinal advection parameter are negative for both the SNPs in the *TYR* and *LCT* loci: the mutation origins are always to the east of the center of mass of the allele frequency distribution seen in present-day data. This perhaps reflects common migratory processes, like the large-scale Neolithic and Bronze Age population movements from east to west, affecting the allele frequencies at these loci across the Eurasian landscape (*Allentoft et al., 2015*; *Haak et al., 2015*). As a form of regularization, we kept the range of explored values for the advection parameters to be small (−2.5–2.5 km per generation), while allowing the diffusion parameters to be explored over a much wider range of values. In certain cases, like the second period of the rs4988235(T) spread when the allele age is assumed to be young (*Appendix 2—table 5*), we find that the advection parameters are fitted at the boundary of the explored range, because the allele needs to spread very fast across the landscape to fit the data.

A future improvement to our method could include other forms of regularization that better account for the joint behavior of the advection and diffusion processes, or the use of priors for these parameters under a Bayesian setting, which could be informed by realistic assumptions about the movement of individuals on a landscape. Bayesian parameter fitting would likely provide a more

robust understanding of the uncertainty of the estimates as well as an opportunity to formally compare different models using Bayes factors, although at the cost of an increase of computational intensity.

When investigating the robustness of the geographic origin of both rs4988235(T) and rs1042602(A), we found that parameters related to the beneficial allele's expansion change in response to different assumed origins of the allele. The resulting allele frequency surface plots, however, appear very similar throughout the later stages of the process, showing that the model tends to adjust the diffusion and advection coefficients in a way such that the allele will end up expanding into the same areas regardless of the origin.

As we apply these methods to longer time scales and broader geographic areas, the assumptions of spatiotemporal homogeneity of the parameters seem less plausible. There may be cases where the allele may have been distributed over a wide geographic area but remained at low frequencies for an extended period of time, complicating the attempts to pinpoint the allele's origin. In our study, we estimated diffusion and selection coefficients separately for two time periods before and after 5000 years ago to account for changes in mobility during the Bronze Age, but this approach may still be hindered by uneven sampling, especially when the allele in question exists at very low frequencies. Notably, our results for the spread of the rs4988235(T) allele during the older time period should be interpreted with caution, since they may be affected by sparse sampling in the early Holocene.

Potential future extensions of our method could incorporate geographic features and historical migration events that create spatially or temporally varying moderators of gene flow. An example of this type of processes is the retreat of glaciers after the last Glacial maximum, which allowed migration of humans into Scandinavia (*Günther et al., 2018*). These changing geographic features could lead to changes in the rate of advection or diffusion across time or space. They could also serve to put more environmentally aware constraints on the geographic origin of the allele, given that it cannot have existed in regions uninhabitable by humans, and to extend our analyses beyond the narrow confines of the Western Eurasian map chosen for this study. One could also envision incorporating variation in population densities over time or known migration processes in the time frames and regions of interest. These might have facilitated rapid, long-range dispersal of beneficial alleles (*Bradburd et al., 2016*; *Hallatschek and Fisher, 2014*) or caused allelic surfing on the wave of range expansions (*Klopfstein et al., 2006*). Additional information like this could come, for example, from previously inferred spatiotemporal demographic processes (e.g. *Racimo et al., 2020b*).

As described above, our model only accounts for diffusion in two directions. Further extension of our model could therefore incorporate anisotropic diffusion (*Othmer et al., 1988*; *Painter and Hillen, 2018*). Another possibility could be the introduction of stochastic process components in order to convert the partial differential equations into stochastic differential equations (*Brown et al., 2000*). Stochastic components could serve to induce spatial autocorrelation and capture local patterns of allele frequency covariance in space that might not be well modeled by the deterministic partial differential equations (PDEs) (*Cressie and Wikle, 2015*). They could also serve to induce stochasticity in allele frequency changes over time as a consequence of genetic drift (*Crow and Kimura, 1970*), allowing one to model the dynamics of more weakly selected variants, where drift plays an important role. Eventually, one could perhaps combine information across loci to jointly model the spatiotemporal frequency surfaces at multiple loci associated with the same trait. This could help clarify the dynamics of polygenic adaptation and negative selection on complex traits (*Irving-Pease et al., 2021*), and perhaps hindcast the genetic value of traits across a landscape.

The availability of hundreds of ancient genomes (*Marciniak and Perry, 2017*) and the increasing interest in spatiotemporal method development (*Bradburd and Ralph, 2019*), such as the one described in this article, will likely lead researchers to posit new questions and hypotheses about the behavior of natural selection. In the case of a beneficial allele spreading on a landscape, new ontologies and vocabulary for describing positive selection in time and space will be needed. Abundant terms exist to classify the initial conditions and dynamics of a selective sweep in a single population (hard sweep, multiple-origin soft sweep, single-origin soft sweep, partial sweep) (*Hermisson and Pennings, 2005*; *Pritchard and Di Rienzo, 2010*; *Hermisson et al., 2017*). In contrast, there is a lack of vocabulary for distinguishing between a scenario of strong selection that is locally constrained in space from a scenario of widespread selection extended over a landscape, or a model of neutral diffusion in space followed by parallel non-neutral increases in frequency at multiple locations. For example, *Ralph and Coop, 2010* showed how multiple localized hard sweeps may be seen as a soft

sweep at a larger population-wide scale. Existing vocabulary for spatiotemporal genetic processes is clearly not enough, limiting the types of questions or hypotheses we can pose about them.

Population genetic models that explicitly account for space and time are an important area of future methodological development (**Bradburd and Ralph, 2019**). We believe that methods such as the one described in this study show great promise at broadening the horizon of our understanding of natural selection across space and time in humans and other species. As in the case of demographic reconstruction (**Ray and Excoffier, 2009**), spatiotemporal information can greatly help improve our knowledge of how natural selection operated in the past.

# Methods
## The model
To describe the allele frequency dynamics in time and space, we first begin by using a deterministic model based on a two-dimensional PDE (**Fisher, 1937**; **Kolmogorov et al., 1937**; **Novembre et al., 2005**). This PDE represents the distribution $p(x, y, t)$ of the allele frequency across a two-dimensional $(x, y)$ landscape at time $t$:

$$\frac{\partial p}{\partial t} = \frac{1}{2}\sigma^2 \frac{\partial^2 p}{\partial x^2} + \frac{1}{2}\sigma^2 \frac{\partial^2 p}{\partial y^2} + \gamma(p, s, d) \tag{1}$$

where

$$\gamma(p, s, d) = p(1 - p)(pd + s(1 - 2p)). \tag{2}$$

Here, $\sigma$ is the diffusion coefficient, $s$ is the selection coefficient, and $d$ is the dominance coefficient (**Novembre et al., 2005**). We assumed an additive model and fixed $d = 2s$ in all analyses below. We call this 'model A,' but we also evaluated the fit of our data under more complex models that are more flexible, and are described below.

Model B is a more general diffusion-reaction model, which incorporates distinct diffusion terms in the longitudinal and latitudinal directions ($\sigma_x$ and $\sigma_y$, respectively):

$$\frac{\partial p}{\partial t} = \frac{1}{2}\sigma_x^2 \frac{\partial^2 p}{\partial x^2} + \frac{1}{2}\sigma_y^2 \frac{\partial^2 p}{\partial y^2} + \gamma(p, s, d) \tag{3}$$

Model C is a generalization of model B that incorporates advection terms in the longitudinal and latitudinal directions (see, e.g. **Cantrell and Cosner, 2004** for a motivation of this type of model in the context of spatial ecology):

$$\frac{\partial p}{\partial t} = \frac{1}{2}\sigma_x^2 \frac{\partial^2 p}{\partial x^2} + \frac{1}{2}\sigma_y^2 \frac{\partial^2 p}{\partial y^2} + v_x \frac{\partial p}{\partial x} + v_y \frac{\partial p}{\partial y} + \gamma(p, s, d) \tag{4}$$

Here, $v_x$ and $v_y$ represent the coefficients for advective velocity along the longitude and latitude respectively.

In Appendix 1, we motivate the construction of these equations using model C as an example and show that *Equation 4* can be obtained by taking an infinitesimal limit of a random walk on a two-dimensional lattice, after including a reaction term due to selection. Models A and B are then shown to be special cases of model C.

For evaluating the likelihood of the observed data, we use a binomial genotype sampling model. Let $g_i \in 0, 1, 2$ be the genotype of individual $i$ at the locus of interest, let $a_i$ be the number of reads carrying ancestral alleles, and let $d_i$ be the number of reads carry derived reads. Let $(x_i, y_i)$ be the coordinates of the location from which individual $i$ was sampled, and $t_i$ its estimated age (e.g., from radiocarbon dating). Then, the likelihood for individual $i$ can be computed as follows:

$$L(d_i, a_i) = \sum_{h=0}^{2} P[d_i, a_i | g_i = h] P[g_i = h | p(x_i, y_i, t_i)] \tag{5}$$

Here, $p(x_i, y_i, t_i)$ is the solution to one of the partial differential equations described above (*Equation 1*, *Equation 2*, or *Equation 4*, depending on the process model chosen), evaluated at location $(x_i, y_i)$ and time $t_i$. In turn, $P[d_i, a_i | g_i = h]$ is the likelihood for genotype $i$. Furthermore, $P[g_i = h | p(x_i, y_i, t_i)]$ is a binomial distribution, where $n$ represents the ploidy level, which in this case is 2:

$$P[g_i = h|p(x_i, y_i, t_i)] = \binom{n}{h} p(x_i, y_i, t_i)^h (1 - p(x_i, y_i, t_i))^{n-h} \tag{6}$$

Then, the likelihood of the entire data can be computed as

$$L(\mathbf{d}, \mathbf{a}) = \prod_{i=1}^{M} L(x_i, y_i, t_i) \tag{7}$$

where M is the total number of individuals for which we have data, d is the vector containing the derived read count for each individual, and a is the vector containing the ancestral read count for each individual. We computed genotype likelihoods directly on the BAM file read data using the SAMtools genotype model (*Li, 2011*) implemented in the software ANGSD (*Korneliussen et al., 2014*).

When only randomly sampled pseudohaploid allele counts are available, we used a Bernoulli sampling likelihood (conditional on the genotype $g_i$) on the right-hand side of *Equation 6* instead. Briefly, assuming that the probability of an individual having genotype $g$ at a particular locus given the underlying allele frequency $p$ follows a binomial distribution and that the probability of sampling a read given the genotype of an individual follows a Bernoulli distribution with probability of success $\frac{1}{2}g$, then the probability of sampling a read given the genotype follows a Bernoulli distribution with probability of success $p$.

## Map

We restricted the geographic area explored by our model fit to be between 30°N to 75°N, and between 10°W and 80°E. For numerical calculations, we used a grid constructed using a resolution of approximately one grid cell per latitude and longitude. We used Harvesine functions in order to transform the distance from degrees to kilometers between two geographic points. The diffusion of the allele frequency was disallowed in the map regions where the topology is negative (i.e., regions under water), based on ETOPO5 data (*NOAA, National Geophysical Data Center, B. C, 1988*). For this reason, we added land bridges between the European mainland and Sardinia, and between the mainland and Great Britain, in order to allow the allele to diffuse in these regions (see *Appendix 2—figure 1*).

## Parameter search

Parameter optimization was done via maximum likelihood estimation with a two-layer optimization set-up. The first layer consists of a simulated annealing approach (*Bélisle, 1992*) starting from 50 random points in the parameter space. The initial 50 points are sampled using Latin hypercube sampling to ensure an even spread across the parameter space. The output of this fit was then fed to the L-BFGS-B algorithm to refine the parameter estimates around the obtained maximum and obtain confidence intervals for the selection, diffusion and advection parameters (*Byrd et al., 1995*).

The parameters optimized were:

- The selection coefficient ($s$), restricted to the range 0.001–0.1.
- Two dispersal parameters $\sigma_x$ and $\sigma_y$ in the longitudinal and latitudinal directions, respectively, restricted to the range of 1–100 square kilometers per generation.
- The longitudinal and latitudinal advection coefficients $vx$ and $vy$, respectively. As a form of regularization, we set the range of explored values to be narrowly centered around zero: –2.5 to 2.5 kilometers per generation.
- The geographic origin of the allele, which is randomly initialized to be any of the 28 spatial points shown in *Appendix 2—figure 2* at the start of the optimization process.

We chose to construct our method in a way that uses the age of the allele as an input parameter rather than estimating it. We do this since there are multiple equally possible solutions with various combinations of allele age and selection coefficient values as shown in *Appendix 2—figure 3*. The latitude and longitude are discretized in our model in order to solve the differential equations numerically, thus the origin of a mutation is measured in terms of discrete units. For this reason, when using the L-BFGS-B algorithm, we fixed the previously estimated origin of the allele and did not explore it during this second optimization layer. For numerical calculations, we used the Livermore Solver for Ordinary Differential Equations (*Hindmarsh, 1983*) implemented in R package 'deSolve' (*Soetaert et al., 2010*), which is a general-purpose solver that can handle both stiff and nonstiff systems. In case of stiff problems, the solver uses a Jacobian matrix. Absorbing boundary conditions were used at the

boundaries of the map. For visualization purposes, we masked the allele frequencies from areas with negative topology (i.e., areas covered by large bodies of water). Time was measured in generations, assuming 29 years per generation. During the optimization, we scaled the time and the parameters by a factor of 10, which allowed us to decrease the execution time of the model.

We initialized the grid by setting the initial allele frequency to be $p_0$ in a grid cell where the allele originates and 0 elsewhere. $p_0$ was calculated as $1/(2*D*A)$, where $D$ is the population density and is equal to 2.5 inhabitants per square kilometer, which is the estimated population density in Europe in 1000 BC (*Colin McEvedy, 1978*; *Novembre et al., 2005*). In the equation, $D$ is multiplied by 2 because we assume that the allele originated in a single chromosome in a diploid individual. $A$ is the area in square kilometers of the grid cell where the allele emerged.

Asymptotic 95% confidence intervals for a given parameter $\theta_j$ were calculated using equation

$$\hat{\theta}_j \pm 1.96\sqrt{(F(\boldsymbol{\theta})^{-1})_{jj}}$$

where $F(\boldsymbol{\theta})$ is an estimate of the observed Fisher information matrix (*Fisher, 1922*; *Efron and Hastie, 2016*; *Casella and Berger, 2001*).

## Implementation

The above-described model was implemented in R version 3.6. To numerically solve the differential equations and obtain maximum likelihood estimates, we used the libraries *deSolve* (*Soetaert et al., 2010*), *ReacTran* (*Soetaert and Meysman, 2012*), and *bbmle* (*Bolker, 2020*). Scripts containing the code used in this article are available on GitHub: https://github.com/RasaMukti/stepadna, (copy archived at swh:1:rev:d024767648d873f329a8e17fcaf6034c99157120; *Muktupavela, 2021*).

## Individual-based simulations

For the individual-based spatiotemporal forward simulations, we first defined a spatial boundary for a population spread across a broad geographic region of Europe. In order to ensure a reasonably uniform distribution of individuals across this spatial range throughout the course of the simulation, we set the maximum distance for spatial competition and mating choice between individuals to 250 km (translated, on a SLiM level, to the interaction parameter *maxDistance*), and the standard deviation of the normal distribution governing the spread of offspring from their parents at 25 km (leveraged in SLiM's *modifyChild*() callback function) (*Haller and Messer, 2019*). We note that we have chosen the values of these parameters merely to ensure a uniform spread of individuals across a simulated landscape. They are not intended to represent realistic estimates for these parameters at any time in human history.

After defining the spatial context of the simulations and ensuring the uniform spread of individuals across their population boundary, we introduced a single beneficial additive mutation in a single individual. In order to test how accurately our model can infer the parameters of interest, we simulated a scenario in which the allele appeared in Central Europe 15,000 years ago with the selection coefficient of the beneficial mutation set to 0.03. Over the course of the simulation, we tracked the position of each individual that ever lived together with its location on a two-dimensional map, as well as its genotype (i.e., zero, one, or two copies of the beneficial allele). We then used this complete information about the spatial distribution of the beneficial allele in each time point to study the accuracy of our model in inferring the parameters of interest.

## Acknowledgements

We thank Graham Gower, Evan Irving-Pease, Montgomery Slatkin, the members of the Racimo group, and two anonymous reviewers for helpful comments and advice. FR and RM were funded by a Villum Fonden Young Investigator award to FR (project no. 00025300). FR was also supported by a Novo Nordisk Fonden Ascending Investigator Award (NF22OC0076816) and a ERC Synergy grant (ID 951385). Additionally, MP and FR were supported by a Lundbeckfonden grant (R302-2018-2155) and a Novo Nordisk Fonden grant (NNF18SA0035006) to the GeoGenetics Centre. TSK was funded by a Carlsberg grant (CF19-0712). JN was funded by NIH grant R01 GM132383.

## Additional information

### Funding

| Funder | Grant reference number | Author |
|---|---|---|
| Villum Fonden | 00025300 | Rasa A Muktupavela Fernando Racimo |
| Lundbeckfonden | R302-2018-2155 | Martin Petr Fernando Racimo |
| Novo Nordisk Fonden | NNF18SA0035006 | Martin Petr Fernando Racimo |
| Carlsbergfondet | CF19-0712 | Thorfinn Korneliussen |
| National Institutes of Health | R01 GM132383 | John Novembre |
| Novo Nordisk Fonden | NNF22OC0076816 | Fernando Racimo |
| European Research Council | 951385 | Fernando Racimo |

The funders had no role in study design, data collection and interpretation, or the decision to submit the work for publication.

### Author contributions

Rasa A Muktupavela, Software, Formal analysis, Visualization, Writing – original draft; Martin Petr, Formal analysis, Writing – original draft, Writing – review and editing; Laure Ségurel, Thorfinn Korneliussen, Supervision, Writing – review and editing; John Novembre, Conceptualization, Supervision, Methodology, Writing – review and editing; Fernando Racimo, Conceptualization, Resources, Supervision, Funding acquisition, Methodology, Writing – original draft, Project administration, Writing – review and editing

### Author ORCIDs

Rasa A Muktupavela http://orcid.org/0000-0001-5590-5948
Martin Petr http://orcid.org/0000-0003-4879-8421
Laure Ségurel http://orcid.org/0000-0001-7339-0976
John Novembre http://orcid.org/0000-0001-5345-0214
Fernando Racimo http://orcid.org/0000-0002-5025-2607

### Decision letter and Author response

Decision letter https://doi.org/10.7554/eLife.73767.sa1
Author response https://doi.org/10.7554/eLife.73767.sa2

---

# Additional files

### Supplementary files

• Transparent reporting form

• Supplementary file 1. Publications that produced data included in the Allen Ancient DNA Resource, compiled by the Reich Lab.

### Data availability

The current manuscript is a computational study, therefore no data have been generated for this manuscript. Software code along with publicly available data used for this study are deposited to GitHub: https://github.com/RasaMukti/stepadna/tree/main/reproducibles (copy archived at swh:1:rev:d024767648d873f329a8e17fcaf6034c99157120).

The following previously published datasets were used:

| Author(s) | Year | Dataset title | Dataset URL | Database and Identifier |
|---|---|---|---|---|
| Reich D, Mallick S | 2019 | Allen Ancient DNA Resource (AADR) | https://reich.hms.harvard.edu/allen-ancient-dna-resource-aadr-downloadable-genotypes-present-day-and-ancient-dna-data | Allen Ancient DNA, AADR |
| Bergström A, McCarthy SA, Hui R, Almarri MA, Ayub Q, Danecek P, Chen Y, Felkel S, Hallast P, Kamm J, Blanché H, Deleuze JF, Cann H, Mallick S, Reich D, Sandhu MS, Skoglund P, Scally A, Xue Y, Durbin R, Tyler-Smith C | 2020 | Human Genome Diversity Panel | http://ftp.1000genomes.ebi.ac.uk/vol1/ftp/data_collections/HGDP/ | EMBL-EBI, HGDP |

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

## Appendix 1

Here, we motivate the construction of model C as a large-scale limit of a random walk model on a lattice (**Karlin and Taylor, 1975**; **Cantrell and Cosner, 2004**). We think of the allele frequency as a variable $p$ that can increase in magnitude due to its inherent advantage (selection), spread across a landscape (diffusion) or move directionally as a consequence of migration (advection). We imagine a lattice composed of small square cells of size $\Delta x \times \Delta y$, where a certain amount of allele frequency $p$ can occur at a given time point $t$. At each small time step (of duration $\Delta t$), inflow and outflow of p can occur in the x-direction with probability h or in the y-direction with probability 1h, and the magnitude of these flows depend on the amount of $p$ present in neighboring cells. If flow of p is along the x-axis, it does so in the positive direction with probability $\alpha$ and in the negative direction with probability $1 - \alpha$. If flow of p is along the y-axis, it does so in the positive direction with probability $\beta$ and in the negative direction with probability $1 - \beta$. The allele frequency can also increase in magnitude locally via a function $\gamma()$ that depends on its dominance (d), selection coefficient (s) and current magnitude ($p(x, y, t)$). Then, we obtain:

$$p(x, y, t + \Delta t) = \gamma(p(x, y, t), s, d)\Delta t + h\alpha p(x - \Delta x, y, t) + h(1 - \alpha)p(x + \Delta x, y, t) + (1 - h)\beta p(x, y - \Delta y, t) +$$
$$(1 - h)(1 - \beta)p(x, y + \Delta, y, t) \tag{8}$$

We can also write this as:

$$p(x, y, t + \Delta t) - p(x, y, t) = h\left(\frac{1}{2} - \alpha\right)[p(x + \Delta x, y, t) - p(x - \Delta x, y, t)] + (1 - h)\left(\frac{1}{2} - \beta\right)[p(x, y + \Delta y, t) -$$
$$p(x, y - \Delta y, t)] + h\frac{1}{2}[p(x + \Delta x, y, t) - 2p(x, y, t) + p(x + \Delta x, y, t)] + (1 - h)\frac{1}{2}[p(x, y + \Delta y, t) - 2p(x, y, t) + \tag{9}$$
$$p(x, y + \Delta y, t)] + \gamma(p(x, y, t), s, d)\Delta t$$

If we divide both sides by $\Delta t$ and take the limit of infinitesimally small $\Delta x$, $\Delta y$, and $\Delta t$, while assuming that, in this limit, $\frac{\Delta x^2}{\Delta t}$ and $\frac{\Delta y^2}{\Delta t}$ are finite (**Okubo, 1980**), we obtain:

$$\frac{\partial p}{\partial t} = \frac{1}{2}h\lambda_x\frac{\partial^2 p}{\partial x^2} + \frac{1}{2}(1 - h)\lambda_y\frac{\partial^2 p}{\partial y^2} + h(1 - 2\alpha)u_x\frac{\partial p}{\partial x} + (1 - h)(1 - 2\beta)u_y\frac{\partial p}{\partial y} + \gamma(p(x, y, t), s, d) \tag{10}$$

where $u_x = \frac{\Delta x}{\Delta t}$, $u_y = \frac{\Delta y}{\Delta t}$, $\lambda_x = \frac{\Delta x^2}{\Delta t}$, $\lambda_y = \frac{\Delta y^2}{\Delta t}$.

If we let $\sigma_x^2 = h\lambda_x$, $\sigma_y^2 = (1 - h)\lambda_y$, $v_x = h(1 - 2\alpha)u_x$, $v_y = (1 - h)(1 - 2\beta)u_y$, then we obtain **Equation 4**. Thus, we can see that the squared diffusion coefficient $\sigma_x^2$ depends on the square of the length of the cells in the x-axis relative to the duration of a time step ($\lambda_x$), and on the probability that flows occurs in the x-axis at a given time step ($h$). Similarly, the squared diffusion coefficient $\sigma_y^2$ depends on the square of the length of the cells in the y-axis relative to the duration of a time step ($\lambda_y$), and on the probability that flows occurs in the y-axis at a given time step ($1 - h$). The advection coefficient $v_x$ depends on the advective velocity along the x-axis ($u_x$) as well as on the probability of flow occurring along the x-axis ($h$) and the directional bias $1 - 2\alpha$, which depends on the probability that flow occurs in the positive x-direction ($\alpha$). Finally, the advection coefficient $v_y$ depends on the advective velocity along the y-axis ($u_y$), as well as on the probability of flow occurring along the y-axis ($1 - h$) and the directional bias $1 - 2\beta$, which depends on the probability that flow occurs in the positive y-direction ($\beta$).

We can recover model B as a special case of model C if we fix $\alpha = \beta = \frac{1}{2}$, assuming isotropy in the two directions, so that $\Delta x = \Delta y$. We can also recover model A if we additionally fix $h = \frac{1}{2}$.

## Appendix 2

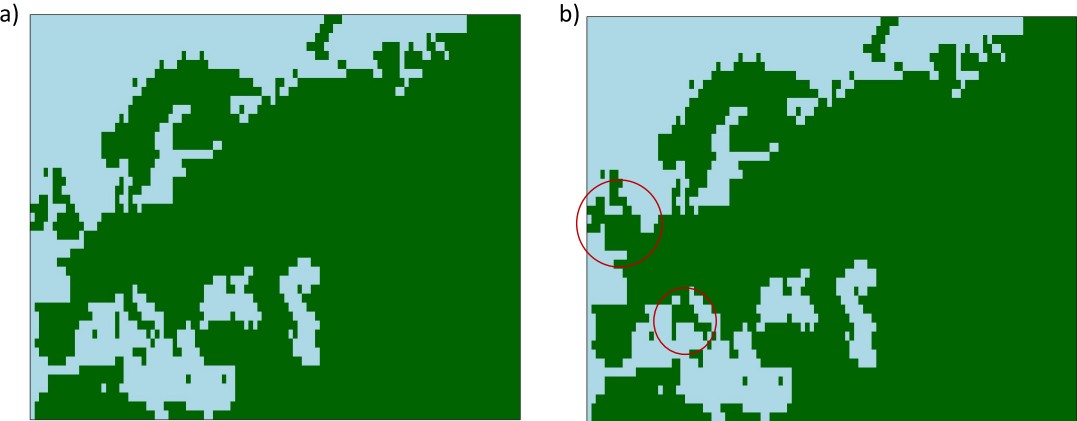

**Appendix 2—figure 1.** Maps showing areas where diffusion in the model is allowed (green) and where it is forbidden (blue). (**a**) Map without land bridges. (**b**) Map containing land bridges indicated with red circles.

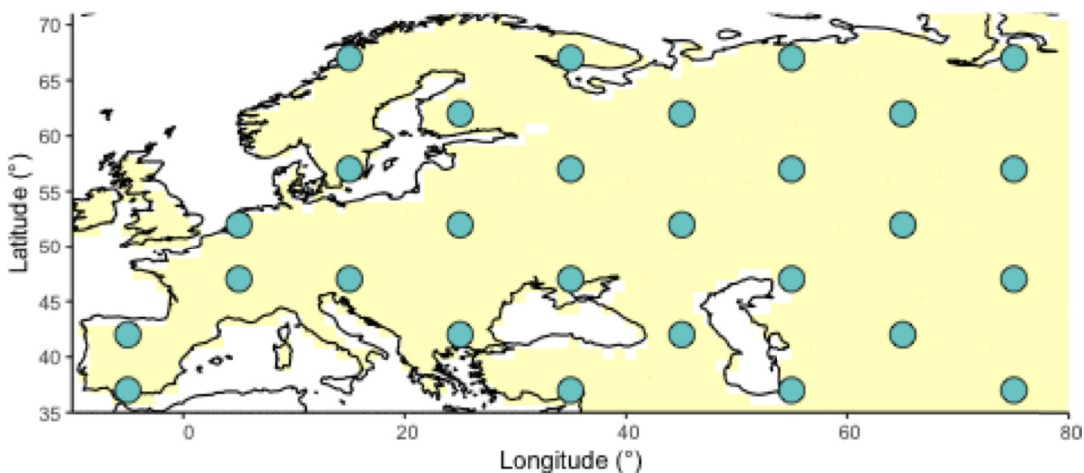

**Appendix 2—figure 2.** Geographic locations for points used as potential origins of the allele at the initialization of the simulated annealing optimization algorithm. Note that, after initialization, the algorithm can continuously explore any points on the map grid that are not necessarily included in this point set.

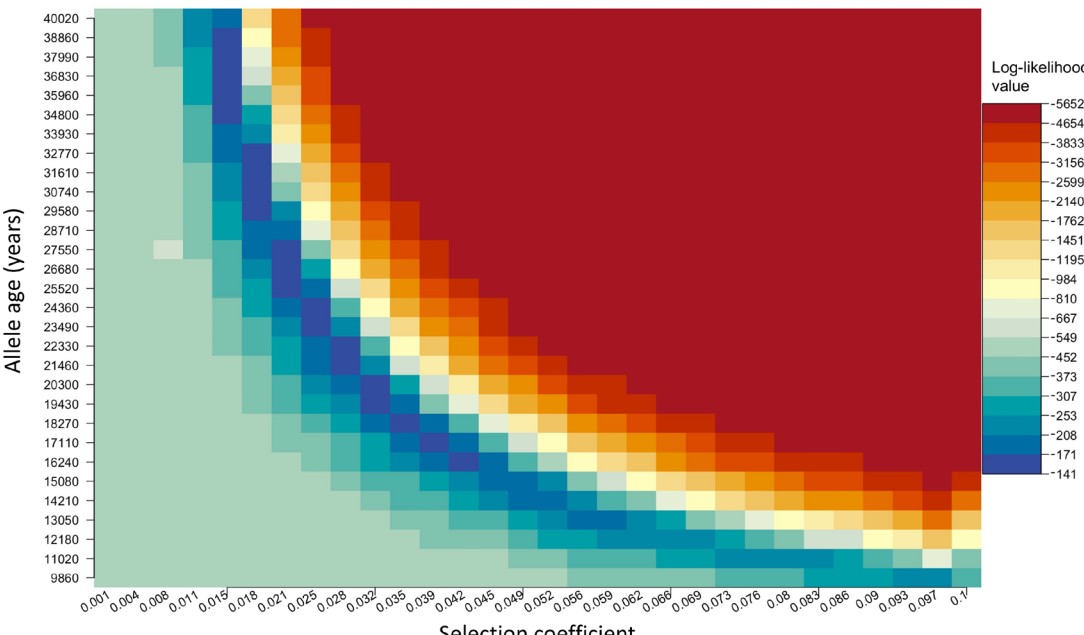

**Appendix 2—figure 3.** Log-likelihood as a function of selection coefficient and age of the allele. Dark blue regions correspond to optimal solutions.

**Appendix 2—table 1.** Parameter values used to generate simulations using numerical solutions to *Equation 3* compared to parameter estimates assuming model B.

The age of the allele was set to 29,000 years in all simulations. Columns named 'long' and 'lat' indicate the longitude and latitude of the geographic origin of the allele, respectively.

| Simulation | | *s* True/pred (95% CI) | $\sigma_x$ (km²/gen) True/pred (95% CI) | $\sigma_y$ (km²/gen) True/pred (95% CI) | Long True/pred | Lat True/pred |
|---|---|---|---|---|---|---|
| | Sample age >5000 | 0.02/0.0192 (0.0187 to 0.0196) | 10/15.244 (2.5042 to 27.9828) | 20/16.963 (11.9263 to 21.9993) | 25/24 | 52/52 |
| B1 | Sample age <5000 | 0.02/0.0027 (0 to 0.0074) | 10/8.805 (0.5631 to 17.0468) | 20/97.432 (97.2566 to 97.6081) | – | – |
| | Sample age >5000 | 0.02/0.0193 (0.0189 to 0.0198) | 10/15.348 (0 to 95.5192) | 30/20.427 (0 to 51.514) | 25/25 | 52/51 |
| B2 | Sample age <5000 | 0.02/0.001 (0 to 0.0059) | 10/10.015 (1.6837 to 18.3472) | 30/100 (99.9933 to 100.0067) | – | – |
| | Sample age >5000 | 0.02/0.0196 (0.0191 to 0.02) | 10/8.149 (6.1551 to 10.143) | 40/49.432 (0 to 135.9428) | 25/24 | 52/51 |
| B3 | Sample age <5000 | 0.02/0.0265 (0.0145 to 0.0386) | 10/7.855 (0 to 19.751) | 40/100 (99.9933 to 100.0067) | – | – |
| | Sample age >5000 | 0.02/0.0188 (0.0188 to 0.0188) | 20/19.037 (19.0311 to 19.0435) | 10/19.254 (19.2439 to 19.2638) | 25/25 | 52/51 |
| B4 | Sample age <5000 | 0.02/0.0142 (0.0111 to 0.0173) | 20/17.354 (4.4083 to 30.2991) | 10/1.489 (0 to 18.5993) | – | – |
| | Sample age >5000 | 0.02/0.0196 (0.019 to 0.0202) | 30/26.409 (11.1997 to 41.6184) | 10/11.429 (7.1825 to 15.6759) | 25/27 | 52/51 |
| B5 | Sample age <5000 | 0.02/0.0215 (0.0183 to 0.0246) | 30/14.3 (0 to 68.176) | 10/1.554 (0 to 16.5985) | – | – |
| | Sample age >5000 | 0.02/0.0199 (0.0192 to 0.0206) | 40/85.415 (41.6058 to 129.2248) | 10/9.02 (7.2853 to 10.7538) | 25/6 | 52/51 |
| B6 | Sample age <5000 | 0.02/0.0163 (0.0112 to 0.0213) | 40/10.403 (0 to 22.533) | 10/22.623 (12.0841 to 33.1614) | – | – |

**Appendix 2—table 2.** Parameter values used to generate simulations using numerical solutions to **Equation 4**, compared to parameter estimates assuming model C.

The age of the allele was set to 29,000 years in all simulations. Columns named 'long' and 'lat' indicate the longitude and latitude of the geographic origin of the allele, respectively.

| Simulation | | $s$ True/pred (95% CI) | $\sigma_x$(km²/gen) True/pred (95% CI) | $\sigma_y$(km²/gen) True/pred (95% CI) | $v_x$ (km/gen) True/pred (95% CI) | $v_y$ (km/gen) True/pred (95% CI) | Long True/pred | Lat True/pred |
|---|---|---|---|---|---|---|---|---|
| | Sample age >5000 | 0.02/0.0189 (0.0188 to 0.0189) | 20/52.246 (52.2051 to 52.2872) | 20/16.373 (16.332 to 16.4139) | -2/-1.675 (-1.6771 to -1.6722) | -2/-2.067 (-2.0702 to -2.0639) | 25/4 | 52/45 |
| C1 | Sample age <5000 | 0.02/0.0231 (0.023 to 0.0233) | 20/11.086 (10.9286 to 11.2441) | 20/15.606 (15.3659 to 15.8467) | -2/0.399 (0.3946 to 0.4037) | -2/-2.491 (-2.5458 to -2.436) | – | – |
| | Sample age >5000 | 0.02/0.0185 (0.0176 to 0.0195) | 10/19.434 (5.6736 to 33.1952) | 20/18.727 (4.8938 to 32.5605) | -1.2/1.579 (1.2671 to 1.8905) | 1.9/-0.801 (-1.1684 to -0.4331) | 25/-6 | 52/38 |
| C2 | Sample age <5000 | 0.02/0.0205 (0.0175–0.0234) | 10/38.144 (10.3123 to 65.9749) | 20/51.094 (14.0489 to 88.1388) | -1.2/-1.299 (-2.7247 to 0.1266) | 1.9/2.493 (2.4929 to 2.4933) | – | – |
| | Sample age >5000 | 0.02/0.0255 (0.0254–0.0256) | 30/59.237 (59.1269 to 59.347) | 10/6.604 (6.5991 to 6.6087) | 1.8/2.195 (2.1918 to 2.1985) | -0.8/0.438 (0.4381 to 0.4387) | 25/65 | 52/66 |
| C3 | Sample age <5000 | 0.02/0.0079 (0–0.0165) | 30/86.511 (0 to 194.0772) | 10/40.693 (24.3946 to 56.9905) | 1.8/-2.498 (-2.4983 to -2.498) | -0.8/-0.014 (-3.4481 to 3.4204) | – | – |
| | Sample age >5000 | 0.02/0.0197 (0.0191–0.0204) | 10/19.647 (14.975 to 24.3197) | 10/13.585 (0 to 27.2936) | 1.2/-0.054 (-0.0968 to -0.0111) | 1/0.72 (0.4278 to 1.0124) | 25/44 | 52/50 |
| C4 | Sample age <5000 | 0.02/0.0137 (0.0046–0.0229) | 10/14.151 (0 to 32.1031) | 10/4.093 (0 to 49.4651) | 1.2/0.8 (-3.4903 to 5.0895) | 1/2.434 (2.4299 to 2.4387) | – | – |

**Appendix 2—table 3.** Parameter value estimates for each of the nine clustering schemes and true parameter values used to generate the deterministic simulation.

The age of the allele was set to 17,400 years. Columns named 'long' and 'lat' indicate the longitude and latitude of the geographic origin of the allele, respectively.

| Sampling scheme | | $S$ (95% CI) | $\sigma_x$(km2/gen) (95% CI) | $\sigma_y$(km2/gen) (95% CI) | $v_x$(km/gen) (95% CI) | $v_y$(km/gen)(95% CI) | Long | Lat |
|---|---|---|---|---|---|---|---|---|
| | Sample age >5000 | 0.0385 (0.0364 to 0.0406) | 24.592 (14.9174 to 34.2675) | 16.194 (4.8309 to 27.5568) | -1.209 (-1.6947 to -0.7229) | -1.555 (-2.0804 to -1.0294) | | |
| Homogeneous 75%/25% | Sample age <5000 | 0.0261 (0.0201 to 0.0321) | 45.725 (19.0071 to 72.4437) | 11.95 (0.0000 to 27.2152) | 2.499 (2.4993 to 2.4996) | -0.905 (-2.3876–0.5783) | 44 | 48 |
| | Sample age >5000 | 0.0379 (0.0364 to 0.0394) | 23.071 (0.0000 to 66.2585) | 11.455 (0.0000 to 26.4585) | -0.827 (-3.2669 to 1.6124) | -0.934 (-1.5199 to -0.3476) | | |
| Homogeneous 50%/50% | Sample age <5000 | 33.944 (0.0292 to 0.0385) | 33.944 (13.1707 to 54.7167) | 6.183 (0.0000 to 13.3805) | 0.478 (-0.5529 to 1.5089) | 0.315 (-0.1828–0.8127) | 46 | 51 |
| | Sample age >5000 | 0.0379 (0.0364 to 0.0394) | 22.619 (14.8534 to 30.3839) | 13.588 (6.1189 to 21.0574) | -1.4 (-1.9213 to -0.8782) | -1.021 (-1.4258 to -0.6161) | | |
| Homogeneous 25%/75% | Sample age <5000 | 0.0322 (0.0257 to 0.0388) | 70.446 (24.6065 to 116.2854) | 3.786 (0.0000 to 21.6379) | 2.499 (2.4984 to 2.4987) | -0.99 (-2.0881 to 0.1079) | 46 | 50 |
| | Sample age >5000 | 0.0378 (0.0378 to 0.0378) | 20.905 (20.904 to 20.9065) | 14.583 (14.5818 to 14.5844) | -1.069 (-1.0687 to -1.0684) | -0.547 (-0.5468 to -0.5467) | | |
| Intermediate 75%/25% | Sample age <5000 | 0.0342 (0.0276 to 0.0407) | 70.405 (29.0665 to 111.7428) | 1.936 (0.0000 to 18.9234) | 2.5 (2.4995 to 2.4998) | -1.865 (-3.0637 to -0.6655) | 44 | 52 |
| | Sample age >5000 | 0.0379 (0.0378–0.0379) | 93.136 (93.0316 to 93.2406) | 10.99 (10.9808 to 10.9994) | 1.103 (1.1009 to 1.1048) | 0.695 (0.6939–0.6954) | | |
| Intermediate 50%/50% | Sample age <5000 | 0.0327 (0.0288–0.0367) | 22.409 (0.0000 to 69.8122) | 18.11 (7.9198 to 28.2994) | 2.496 (2.4962 to 2.4965) | -2.499 (-2.4992 to -2.4989) | 34 | 57 |

*Appendix 2—table 3 Continued on next page*

*Appendix 2—table 3 Continued*

| Sampling scheme | | $S$ (95% CI) | $\sigma_x$(km2/gen) (95% CI) | $\sigma_y$(km2/gen) (95% CI) | $v_x$(km/gen) (95% CI) | $v_y$(km/gen)(95% CI) | Long | Lat |
|---|---|---|---|---|---|---|---|---|
| | Sample age >5000 | 0.0386 (0.0371–0.0402) | 21.385 (14.0301 to 28.7407) | 12.335 (3.3756 to 21.2943) | -1.028 (-1.4606 to -0.5956) | -1.307 (-1.7696 to -0.845) | | |
| Intermediate 25%/75% | Sample age <5000 | 0.0295 (0.026–0.0329) | 21.197 (6.0797 to 36.3142) | 11.318 (2.651 to 19.9851) | 2.5 (2.4997 to 2.5000) | -0.757 (-1.391 to -0.123) | 43 | 49 |
| | Sample age >5000 | 0.0362 (0.0336–0.0389) | 33.07 (0.0000 to 78.1418) | 26.744 (0.0000 to 155.2413) | -0.087 (-0.3579 to 0.1832) | -2.001 (-4.7547–0.7524) | | |
| Extreme 75%/25% | Sample age <5000 | 0.0299 (0.0266–0.0332) | 16.702 (0.0000 to 40.5995) | 3.048 (0.0000 to 13.034) | 2.197 (0.8463 to 3.5479) | -2.499 (-2.4995 to -2.4992) | 39 | 46 |
| | Sample age >5000 | 0.0392 (0.0369–0.0416) | 95.472 (95.2997 to 95.6441) | 11.22 (5.3235 to 17.1167) | 1.633 (-2.5434 to 5.8102) | 0.258 (0.0818–0.434) | | |
| Extreme 50%/50% | Sample age <5000 | 0.0355 (0.0314–0.0396) | 11.756 (10.3763 to 13.1361) | 11.817 (10.1474 to 13.4863) | 2.5 (2.2069 to 2.7928) | -0.362 (-0.4325 to -0.2919) | 36 | 57 |
| | Sample age >5000 | 0.047 (0.047–0.047) | 7.909 (7.9075 to 7.9106) | 5.941 (5.9403 to 5.942) | 0.454 (0.4537 to 0.4538) | -2.273 (-2.2732 to -2.2729) | | |
| Extreme 25%/75% | Sample age <5000 | 0.0434 (0.0385–0.0483) | 40.097 (33.7903 to 46.4034) | 12.118 (5.781 to 18.4555) | 2.5 (1.6706 to 3.3285) | -1.435 (-2.3958 to -0.4736) | 34 | 38 |
| | True parameter values | 0.04 | 25 | 10 | 1.8 | -0.8 | 25 | 52 |

**Appendix 2—table 4.** Parameter values estimated using model C for the forward simulation created using SLiM.

Columns named 'long' and 'lat'indicate the longitude and latitude of the geographic origin of the allele, respectively.

| $s$(95% CI) | $\sigma_x$(km²/gen) (95% CI) | $\sigma_y$(km²/gen) | $v_x$(km/gen)(95% CI) | $v_y$(km/gen)(95% CI) | Long | Lat | Allele age (years) |
|---|---|---|---|---|---|---|---|
| 0.0366 (0.0357 to 0.0375) | 58.583 (49.1983 to 67.9669) | 63.733 (3.6601 to 123.8056) | -0.436 (-0.8077to -0.0649) | -1.564 (-3.0915 to -0.0355) | 15 | 47 | 15,000 |

**Appendix 2—table 5.** Summary of parameter estimates for rs4988235(T).

The upper two rows correspond to results obtained assuming the allele age to be the point estimate of the start of selection onset from *Itan et al., 2009*: 7441 years ago. The middle two rows and the bottom two rows show results assuming the age to be either the lower or the higher ends of the age's 95% confidence interval from *Itan et al., 2009*. Columns named 'long' and 'Lat' indicate the longitude and latitude of the geographic origin of the allele, respectively.

| | $s$(95% CI) | $\sigma_x$(km²/gen) (95% CI) | $\sigma_y$(km²/gen) (95% CI) | $v_x$ (km/gen) (95% CI) | $v_y$ (km/gen) (95% CI) | Long | Lat | Allele age (years) |
|---|---|---|---|---|---|---|---|---|
| Sample age >5000 | 0.0993 (0.0993 to 0.0993) | 20.293 (15.5643 to 25.0226) | 15.642 (9.9963 to 21.2871) | -0.575 (-0.8055 to -0.3446) | 0.435 (0.319–0.5512) | | | |
| Sample age <5000 | 0.0328 (0.0327 to 0.0329) | 94.901 (94.2585 to 95.5435) | 85.612 (84.6975 to 86.526) | -1.211 (-1.2197 to -1.2019) | -2.5 (-2.5136 to -2.4855) | 43 | 51 | 7441 |
| Sample age >5000 | 0.0867 (0.0866 to 0.0867) | 24.27 (24.2658 to 24.2734) | 28.328 (28.3234 to 28.3326) | -0.398 (-0.3985 to -0.3984) | -2.055 (-2.0562 to -2.0547) | | | |
| Sample age <5000 | 0.0321 (0.0319–0.0323) | 97.325 (97.1434–97.5061) | 87.416 (85.6745 to 89.1578) | -2.5 (-2.5 to -2.4997) | -2.389 (-2.3935 to -2.3845) | 35 | 46 | 8683 |
| Sample age >5000 | 0.0994 (0.0994–0.0994) | 22.92 (15.0004–30.8397) | 17.884 (13.8709 to 21.8967) | 0.327 (0.1726 to 0.4818) | -0.295 (-0.3678 to -0.2229) | | | |
| Sample age <5000 | 0.0572 (0.057–0.0574) | 95.014 (93.6242–96.4032) | 85.249 (82.9662 to 87.5322) | -2.499 (-2.4992 to -2.4989) | -1.679 (-1.7919 to -1.5658) | 35 | 49 | 6256 |

**Appendix 2—table 6.** Parameter estimates for rs4988235(T) using the allele age inferred in *Albers and McVean, 2020*.

Columns named 'long' and 'lat' indicate the longitude and latitude of the geographic origin of the allele, respectively.

| | S(95% CI) | $\sigma_x$(km²/gen) (95% CI) | $\sigma_y$(km²/gen) (95% CI) | $v_x$ (km/gen) (95% CI) | $v_y$ (km/gen) (95% CI) | Long | Lat | Allele age (years) |
|---|---|---|---|---|---|---|---|---|
| Sample age >5000 | 0.0285 (0.0285 to 0.0285) | 1.25 (1.2492 to 1.25) | 44.619 (44.5944 to 44.6445) | -0.177 (-0.1773 to -0.1771) | 1.925 (1.9247 to 1.9262) | | | |
| Sample age <5000 | 0.0255 (0.0252 to 0.0258) | 92.545 (91.6963 to 93.3941) | 87.545 (85.3525 to 89.7369) | -2.499 (-2.4992 to -2.4989) | -2.271 (-2.4127 to -2.1297) | 32 | 66 | 20,106 |

**Appendix 2—table 7.** Summary of parameter estimates for rs1042602(A).

Upper two rows correspond to model fit when allele age is set to be the point estimate *Albers and McVean, 2020*: 26,367 years ago. The middle two rows and the bottom two rows show results assuming the age to be either the lower or the higher ends of the allele age's 95% confidence interval from *Albers and McVean, 2020*. Columns named 'long' and 'lat' indicate the longitude and latitude of the geographic origin of the allele, respectively.

| | S (95% CI) | $\sigma_x$ (km²/gen) (95% CI) | $\sigma_y$ (km²/gen) (95% CI) | $v_x$(km/gen) (95% CI) | $v_y$(km/gen) (95% CI) | Long | Lat | Allele age (years) |
|---|---|---|---|---|---|---|---|---|
| Sample age >5000 | 0.0221 (0.0216 to 0.0227) | 71.668 (24.7274 to 118.6092) | 50.434 (25.6535 to 75.2136) | -2.268 (-3.006 to -1.5304) | -0.486 (-0.8661 to -0.1053) | | | |
| Sample age <5000 | 0.0102 (0.0083 to 0.012) | 69.25 (14.0247 to 124.4756) | 95.281 (95.1087 to 95.453) | 0.849 (-0.0783 to 1.7769) | -0.503 (-0.929 to -0.076) | 44 | 43 | 26,367 |
| Sample age >5000 | 0.0214 (0.0205 to 0.0223) | 57.914 (0 to 131.3177) | 83.846 (0 to 246.6688) | -2.111 (-2.8784 to -1.3429) | 1.305 (-0.8411 to 3.4519) | | | |
| Sample age <5000 | 0.01 (0.0078 to 0.0121) | 88.218 (0 to 190.105) | 96.216 (96.0422 to 96.3898) | 1.19 (-0.7489 to 3.1293) | -0.88 (-2.0897 to 0.3299) | 46 | 51 | 27,315 |
| Sample age >5000 | 0.023 (0.023 to 0.0231) | 75.857 (75.8065 to 75.9071) | 48.992 (48.9166 to 49.0674) | -2.362 (-2.3655 to -2.3593) | -0.837 (-0.8371 to -0.8362) | | | |
| Sample age <5000 | 0.0099 (0.0085 to 0.0112) | 72.847 (67.7991 to 77.8949) | 92.867 (75.4925 to 110.2412) | 0.497 (0.2717 to 0.7214) | -0.685 (-0.8076 to -0.5628) | 43 | 42 | 25,424 |

**Appendix 2—table 8.** Summary of parameter estimates for rs4988235(T) when the origin of the allele is forced to be at different points in the map (top panel corresponds to the original fit for the geographic position).

In all cases, the estimated age of allele that was inputted into the model is 7441 years ago. Columns named 'long' and 'lat' indicate the longitude and latitude of the geographic origin of the allele, respectively.

| | S(95% CI) | $\sigma_x$(km²/gen) (95% CI) | $\sigma_y$(km²/gen) (95% CI) | $v_x$ (km/gen) (95% CI) | $v_y$ (km/gen) (95% CI) | Long | Lat |
|---|---|---|---|---|---|---|---|
| Sample age >5000 | 0.0993 (0.0993 to 0.0993) | 20.293 (15.5643 to 25.0226) | 15.642 (9.9963 to 21.2871) | -0.575 (-0.8055 to -0.3446) | 0.435 (0.319 to 0.5512) | | |
| Sample age <5000 | 0.0328 (0.0327 to 0.0329) | 94.901 (94.2585 to 95.5435) | 85.612 (84.6975 to 86.526) | -1.211 (-1.2197 to -1.2019) | -2.5 (-2.5136 to -2.4855) | 43 | 51 |
| Sample age >5000 | 0.0985 (0.0985 to 0.0985) | 3.103 (3.1027 to 3.1031) | 44.876 (44.8747 to 44.8768) | 0.354 (0.3537–0.3537) | -0.663 (-0.6634 to -0.6633) | | |
| Sample age <5000 | 0.0413 (0.0411 to 0.0415) | 96.029 (95.8493 to 96.2087) | 85.711 (83.6634 to 87.7594) | -2.5 (-2.5002 to -2.4998) | -1.318 (-1.46 to -1.1764) | 33 | 51 |
| Sample age >5000 | 0.0979 (0.0978 to 0.0979) | 70.388 (70.3697 to 70.4065) | 2.628 (2.6271 to 2.6286) | -2.328 (-2.3286 to -2.3276) | 1.216 (1.2159–1.2164) | | |
| Sample age <5000 | 0.0376 (0.0374 to 0.0377) | 3.705 (1.9497 to 5.4607) | 77.019 (74.9065– to 79.1311) | -2.413 (-2.4174 to -2.4084) | -2.5 (-2.4999 to -2.4995) | 53 | 51 |
| Sample age >5000 | 0.0991 (0.0991 to 0.0992) | 1.218 (1.218 to 1.2183) | 15.127 (15.1256 to 15.1287) | -0.781 (-0.781 to -0.7807) | 2.452 (2.452 to 2.4526) | | |
| Sample age <5000 | 0.0359 (0.0357 to 0.0361) | 96.836 (96.6538 to 97.0183) | 86.616 (83.9434 to 89.2891) | -2.499 (-2.4994 to -2.499) | -2.219 (-2.3368 to -2.1009) | 43 | 61 |
| Sample age >5000 | 0.0999 (0.0999 to 0.0999) | 27.442 (27.4385 to 27.4464) | 11.879 (11.8781 to 11.8801) | -1.582 (-1.5824 to -1.582) | -1.638 (-1.6382 to -1.638) | | |
| Sample age <5000 | 0.0355 (0.0353 to 0.0357) | 97.044 (96.8637 to 97.2236) | 86.223 (83.4533 to 88.992) | -2.499 (-2.4996 to -2.4992) | -2.148 (-2.2811 to -2.0141) | 43 | 41 |

**Appendix 2—table 9.** Parameter estimates for rs4988235(T) using the geographic origin of the allele inferred in **Itan et al., 2009**.

Columns named 'long' and 'lat' indicate the longitude and latitude of the geographic origin of the allele, respectively.

| | $S$(95% CI) | $\sigma_x$(km²/gen) (95% CI) | $\sigma_y$(km²/gen) (95% CI) | $v_x$ (km/gen) (95% CI) | $v_y$ (km/gen) (95% CI) | Long | Lat | Allele age (years) |
|---|---|---|---|---|---|---|---|---|
| Sample age >5000 | 0.0989 (0.0989 to 0.0989) | 9.341 (9.3402 to 9.341) | 3.264 (3.2635 to 3.2643) | 2.338 (2.3379 to 2.3381) | -0.21 (-0.2098 to -0.2098) | | | |
| Sample age <5000 | 0.0358 (0.0357 to 0.036) | 97.086 (96.9059 to 97.2657) | 87.043 (85.1968 to 88.8895) | -2.434 (-2.4385 to -2.4294) | -2.499 (-2.4994 to -2.499) | 13 | 48 | 7441 |

**Appendix 2—table 10.** Summary of parameter estimates for rs1042602(A) when the origin of the allele is forced to be at different points in the map (top panel corresponds to the original fit for the geographic position).

In all cases, the estimated age of allele that was inputted into the model is 26,367 years ago.

Columns named 'long' and 'lat' indicate the longitude and latitude of the geographic origin of the allele, respectively.

| | $S$(95% CI) | $\sigma_x$(km²/gen) (95% CI) | $\sigma_y$(km²/gen) (95% CI) | $v_x$ (km/gen) (95% CI) | $v_y$ (km/gen) (95% CI) | Long | Lat |
|---|---|---|---|---|---|---|---|
| Sample age >5000 | 0.0221 (0.0216 to 0.0227) | 71.668 (24.7274 to 118.6092) | 50.434 (25.6535 to 75.2136) | -2.268 (-3.006 to -1.5304) | -0.486 (-0.8661 to -0.1053) | | |
| Sample age <5000 | 0.0102 (0.0083 to 0.012) | 69.25 (14.0247 to 124.4756) | 95.281 (95.1087 to 95.453) | 0.849 (-0.0783 to 1.7769) | -0.503 (-0.929 to -0.076) | 44 | 43 |
| Sample age >5000 | 0.0227 (0.0223 to 0.0231) | 42.745 (33.6354 to 51.8541) | 96.993 (96.8183 to 97.1683) | -2.437 (-2.4412 to -2.4324) | -0.266 (-0.4848 to -0.0468) | | |
| Sample age <5000 | 0.0095 (0.007 to 0.0119) | 93.477 (7.6582 to 179.2965) | 99.634 (0 to 205.4586) | -2.499 (-3.2101 to -1.7873) | 2.057 (-0.7888 to 4.903) | 54 | 43 |
| Sample age >5000 | 0.0221 (0.0221 to 0.0221) | 47.691 (47.6686 to 47.7127) | 71.367 (71.336 to 71.3986) | -2.164 (-2.1652 to -2.1637) | 1.839 (1.8387 to 1.8392) | | |
| Sample age <5000 | 0.0112 (0.0093 to 0.0131) | 87.959 (0 to 215.8939) | 88.951 (25.5422 to 152.3589) | 2.108 (-0.2061 to 4.4227) | -2.237 (-5.7828 to 1.3083) | 44 | 53 |
| Sample age >5000 | 0.0219 (0.0209 to 0.0229) | 73.106 (38.1699 to 108.043) | 76.835 (24.0025 to 129.6684) | -2.429 (-2.4335 to -2.4248) | -1.474 (-2.8769 to -0.0706) | | |
| Sample age <5000 | 0.0102 (0.0083 to 0.0121) | 88.216 (0 to 192.1057) | 95.401 (95.2283 to 95.573) | 0.871 (-0.2474 to 1.9893) | -1.026 (-2.6161 to 0.564) | 44 | 33 |

