## [Editor Report]

This is an important manuscript that presents an elegant framework to infer the dynamics of beneficial alleles over time and space. The authors present a new method and show, convincingly, its utility and great potential to reconstruct the evolutionary history of beneficial alleles. The method is also applied to loci that likely mediated human genetic adaptations, contributing to our understanding of human recent evolution. The work will be of broad interest to evolutionary biologists who seek to understand the dynamics of beneficial mutations in populations.

---

## [Decision Letter]

**Decision letter after peer review:**

Thank you for submitting your article "Modelling the spatiotemporal spread of beneficial alleles using ancient genomes" for consideration by *eLife*. Your article has been reviewed by 2 peer reviewers, and the evaluation has been overseen by a Reviewing Editor and George Perry as the Senior Editor. The following individuals involved in review of your submission have agreed to reveal their identity: Isabel Alves (Reviewer #1); Anders Eriksson (Reviewer #2).

Essential revisions:

The reviewers and editor found the manuscript elegant and interesting and appreciate the novelty and promise of your new method. They also value your new inferences on two important targets of positive selection in humans. You will find the individual reviews below, to be addressed individually. Here we provide a summary of the desired revisions, to assist you in the revision process.

1) We believe that the manuscript needs a more systematic test for the sensitivity of the method to 1) misspecification of the age of the aDNA samples, and 2) the geographic location/clustering of the samples. This could be done for example by exploring the effect of these factors on the accuracy of the parameter estimates.

2) Related, we believe that further discussion is needed on the extent to which differences observed across time periods (e.g. before and after 5000 BP) reflect particularities of the alleles (e.g. selection), populations (e.g mobility) or the density of the samples across time and space.

3) Where models are applied to specific loci, results are often reported in the text without enough quantitative information and we think that clearer and more detailed information about the inferences must be presented in the text. As an example, where the model is run for different allele age estimates the (approximate maximum) likelihoods of parameter estimates for the different scenarios should be clearly shown in the text.

4) We believe that the authors should clarify some of their choices for the method. We would like to see a discussion of whether the method may be suitable for treating allele age as a free parameter, or whether it is possible (and superior) to report Bayesian posterior distributions for the parameters and Bayes factors to compare the fit of different models to the data.

5) The loci studied represent unusual cases of very strong positive selection in particular geographic areas. We would welcome more discussion regarding the extent to which the current version of the method can be applied to other alleles, to other geographic regions and even to other species.

6) Finally, the reviewers raised issues regarding the presentation of the numerical implementation that we hope that the authors can easily address.

*Reviewer #1 (Recommendations for the authors):*

I very much appreciated reading this manuscript and the effort of the authors in developing such an exciting tool. However, I still have some concerns about the accuracy and the meaning of the estimates.

1) We know that ancient DNA samples are clustered in time and space because they are an aggregate of samples coming from different studies often with different scopes in terms of time and geographical area covered. What's the impact of having clustered samples in the parameter estimates? The authors comment on that a little bit when comparing the accuracy tests performed with the deterministic vs forward in time simulations. However, I think a more systematic analysis of different sampling schemes would help clarifying the robustness of the parameter inference. Moreover, I think this would greatly help understanding the differences in behavior of the statistical method before and after 5000 BP. Indeed the authors allow the model parameters to differ across different time periods (before and after 5000 BP) but it is not clear whether differences in the estimates reflect any particularity of the populations (e.g mobility) or they just reflect differences in the density of the samples across time and space.

2) How would the method perform if simulations would be performed without advection (numerical solution to equation 3) but parameter estimates would be inferred from a model C, which includes advection ?

3) Finally and more of a general question: what's the point of estimating the geographical origin of an allele if there are no samples dated from around the same time as the assumed allele age? I have the feeling this can be easily misinterpreted but maybe I am missing something. The authors discuss this (lines 199-210) when discussing the results obtained for the dynamics of the rs4988235(T) for the two assumed allele ages but it is not clear to me if the estimate for the local of origin is actually meaningful under similar situations. I wonder if with simulation one could try to understand what the method is retrieving…

With respect to the presentation of the work I found that the manuscript is globally well explained but there are some parts a bit difficult to follow:

1) it should be clearly stated in the main that the age of the allele is provided as input and not inferred contrarily to the geographical origin of the allele and the value used to perform the inference should also be provided in the corresponding part of the main text.

2) The comparison of the results for the two ages assumed for the rs4988235(T) mutation (Itan et al. 2009 and Albers and McVean 2020), mainly between lines 199-210, is very confusing. From Figure 5a it seems there is no information on the allele for the time period between the allele age inferred by Itan et al. 2009 and Albers and McVean 2020. However, from the text it is not clearly stated whether there is or not information. If there is no information at all how does the algorithm infer an origin in Northeastern Europe?

3) It would also help to follow the results if the authors would clearly state in the main text (lines: 212-230) the age of the allele assumed for the rs1042602(A).

*Reviewer #2 (Recommendations for the authors):*

Given the ability to of the model to sample at any point in space and time, it would have been very interesting to test the model under actual geographical and temporal distributions of data, such as the Allen Ancient DNA Resource used for one of the empirical investigations, perhaps even taking the observed pattern of missingness in the data into account.

I would recommend using latitude and longitude systematically throughout the manuscript instead of mixing x, y, latitude, longitude.

In the methods: Please fix the typo in Eq (9). Also please clarify which boundary conditions where used, and how the equations were solved numerically (using Eq. (8) or (9)?), including how boundary conditions were implemented.

L437: " the left-hand side of equation 5". This doesn't look right, do you mean the right-hand side of Eq. (6)?

[Editors' note: further revisions were suggested prior to acceptance, as described below.]

*Reviewer #1 (Recommendations for the authors):*

It is now the second time I review the manuscript "Modelling the spatiotemporal spread of beneficial alleles using ancient genomes" by Muktupavela et al. and I am glad to see that the authors addressed most of the reviewers' concerns. There is now a more systemic assessment of the advantages/limitations of the method which I truly believe to be crucial for future implementations. That being said, I would like to recommend this manuscript for publication.

*Reviewer #2 (Recommendations for the authors):*

The authors have made substantial improvements to the manuscript, and is ready for acceptance except for one remaining point:

Contrary to what the authors claim in their rebuttal, Eqs (8) and (9) in Appendix 1 still contain errors.

First, p(x+\δ x,y,t) and p(x+\δ x,y,t) appears twice on line 827.

Second, the coefficients of p(x,y,t) on both sides of Eq (9) sum to -1, which is correct. The term should therefore be absent (have coefficient zero) in Eq (8).

---

## [Author Response]

Reviewer #1 (Recommendations for the authors):I very much appreciated reading this manuscript and the effort of the authors in developing such an exciting tool. However, I still have some concerns about the accuracy and the meaning of the estimates.1) We know that ancient DNA samples are clustered in time and space because they are an aggregate of samples coming from different studies often with different scopes in terms of time and geographical area covered. What's the impact of having clustered samples in the parameter estimates? The authors comment on that a little bit when comparing the accuracy tests performed with the deterministic vs forward in time simulations. However, I think a more systematic analysis of different sampling schemes would help clarifying the robustness of the parameter inference. Moreover, I think this would greatly help understanding the differences in behavior of the statistical method before and after 5000 BP. Indeed the authors allow the model parameters to differ across different time periods (before and after 5000 BP) but it is not clear whether differences in the estimates reflect any particularity of the populations (e.g mobility) or they just reflect differences in the density of the samples across time and space.

We thank the reviewer for highlighting the importance of taking into account clustering effects that could potentially arise by aggregating aDNA data from studies with different sampling schemes. To elaborate on this issue in more detail, we have included a new analysis (chapter “Impact of sample clustering on parameter estimates” in the manuscript) of the expected impact of different sampling and clustering schemes on our inferences. We used a deterministic simulation to create three different degrees of clustering which we will refer to as “homogeneous”, “intermediate” or “extreme” by varying the area from which we sample individuals to be used in our inferences, now included as Figure 3—figure supplement 1 in the paper. Additionally, we also tested the impact of biased temporal sampling in the periods before and after 5000 year BP by oversampling in the ancient period (75%/25%), equal sampling in the two periods (50%/50%), and oversampling in the recent period (25%/75%). Because we evaluated this temporal bias for each of the three spatial clustering sampling scenarios, this resulted in a total of 9 different sampling scenarios. We note that the third “extreme” spatial clustering scenario is completely unrealistic and one would not expect inferences of any degree of accuracy from it, but we believe it gives a good idea of the behaviour of our method in the limit case of extremely restricted spatial sampling.

A comparison of allele frequency maps generated using true parameter values and using parameter estimates from the different sampling schemes are shown in the manuscript Figure 3—figure supplements 2-9. In Figure 3—figure supplement 6 and Figure 3 in the manuscript we show the allele frequency map generated using the “intermediate 75%/25%” clustering scheme. Parameter estimates used to generate all these figures are summarised in Appendix 2–Table 3 in the manuscript. Overall we can see that the allele frequency maps inferred from these scenarios closely resemble the maps generated using the true parameter values, despite the challenges in finding accurate values for the individual point estimates of some of the parameters, highlighting that various combinations of diffusion and advection coefficients can produce similar underlying frequency maps (as discussed in the manuscript section “Performance on deterministic simulations”). This suggests that the joint spatiotemporal information encoded in the inferred maps (not just the individual parameters estimates) should be used in interpreting model outputs, particularly when it comes to the advection and diffusion parameters. The selection coefficient estimates are inferred highly accurately, regardless of the sampling scheme chosen, and lie close to the true value, with only a slight underestimation in the time period after 5000 years BP (with the exception of “extreme 25%/75%”).

2) How would the method perform if simulations would be performed without advection (numerical solution to equation 3) but parameter estimates would be inferred from a model C, which includes advection ?

The following text was added to the main text as section “Advection model application to non-advection simulations”.

“We assessed the performance of the model C, which includes advection coefficient estimates on simulations generated without advection (results shown in Figure 1—figure supplement 6 and Figure 1—figure supplement 7). We can observe that the advection coefficients are inferred to be non-zero (Figure 1—figure supplement 6b and Figure 1—figure supplement 7b), however the inferred maps of spatial allele frequency dynamics closely resemble the ones obtained with true parameter values (Figure 1—figure supplement 6a and Figure 1—figure supplement 7a). This shows that complex interactions between the diffusion and advection coefficients can result in similar outcomes even when only diffusion is considered in the model.

The inference of the origin of the allele also differs when we compare the results for using model B and model C. In order to understand better how the model estimates the allele origin, we highlighted the first individual in simulations B1 and B4 that carries the derived allele. We can see that in the case of simulation B1 the inferred origin of the allele is close to the first observance of the derived allele in the model which includes advection. In contrast, when the advection is not included, the origin of the allele is inferred to be closer to where it is initially rising in frequency (Figure 1—figure supplement 1a and Figure 1—figure supplement 4a). However, this is not always the case. For instance, if we look at the results from the advection model on simulation B4, we can see that the origin of the allele is inferred relatively far from the sample known to have carried the first instance of the derived allele. Therefore, if there is a relatively large interval between the time when the allele originated and when the first ancient genomes are available, the beneficial allele can spread widely, but as this spread is not captured by any of the data points, inference of the precise origin of the selected allele is nearly impossible.”

3) Finally and more of a general question: what's the point of estimating the geographical origin of an allele if there are no samples dated from around the same time as the assumed allele age? I have the feeling this can be easily misinterpreted but maybe I am missing something. The authors discuss this (lines 199-210) when discussing the results obtained for the dynamics of the rs4988235(T) for the two assumed allele ages but it is not clear to me if the estimate for the local of origin is actually meaningful under similar situations. I wonder if with simulation one could try to understand what the method is retrieving…

Indeed, inference about spatio-temporal dynamics of an allele in a time period which is not covered by any ancient samples whatsoever makes little sense beyond making extremely broad conclusions (such as, perhaps, to get an idea where it is very unlikely to have originated). Specifically, our new results described above show that in situations when the advection is disallowed in the model, the allele origin tends to be inferred to lie within the region of highest allele frequencies. When the advection is considered, the model infers the origin of the allele to be in close proximity to the very first carrier of the derived in the data set of available sequenced genomes. In other words, in the absence of any useful information, difficulties may arise when inferring this parameter when the age of the allele is much larger than the age of the oldest sample in the data set.

With respect to the presentation of the work I found that the manuscript is globally well explained but there are some parts a bit difficult to follow:1) it should be clearly stated in the main that the age of the allele is provided as input and not inferred contrarily to the geographical origin of the allele and the value used to perform the inference should also be provided in the corresponding part of the main text.

On lines 124-128 we mention that we use previously published allele age estimates as starting points for the diffusion process.

We mention the exact age estimates of the rs4988235(T) allele that we use as input on lines 239-240 under the section “Dynamics of the rs4988235(T) allele”.

The allele age estimate for the rs1042602(A) allele has now been added on line 275. Thank you for bringing up this issue.

2) The comparison of the results for the two ages assumed for the rs4988235(T) mutation (Itan et al. 2009 and Albers and McVean 2020), mainly between lines 199-210, is very confusing. From Figure 5a it seems there is no information on the allele for the time period between the allele age inferred by Itan et al. 2009 and Albers and McVean 2020. However, from the text it is not clearly stated whether there is or not information. If there is no information at all how does the algorithm infer an origin in Northeastern Europe?

We rephrased lines 252-260. We hope that the information regarding the inference of the Northeastern European origin is now conveyed more clearly.

“Assuming the older age estimate from Albers and McVean (2020), the origin of the allele is estimated to be in the Northeast of Europe (Figure 6—figure supplement 1), which is at a much higher latitude than the first occurrence of the allele, in Ukraine. Due to the deterministic nature of the model, the frequency is implicitly imposed to expand in a region where there are no actual observed instances of the allele. The model compensates for this by placing the origin in an area with a lower density of available aDNA data and thus avoiding an overlap of the increasing allele frequencies with individuals who do not carry the derived rs4988235(T) allele (see Figure 5a). As the model expands rapidly in the southern direction (Appendix 2–Table 6) it eventually reaches the sample carrying the derived variant in Ukraine.”

3) It would also help to follow the results if the authors would clearly state in the main text (lines: 212-230) the age of the allele assumed for the rs1042602(A).

We have now added in the main text the age of the rs1042602(A) allele in the line 275.

Reviewer #2 (Recommendations for the authors):Given the ability to of the model to sample at any point in space and time, it would have been very interesting to test the model under actual geographical and temporal distributions of data, such as the Allen Ancient DNA Resource used for one of the empirical investigations, perhaps even taking the observed pattern of missingness in the data into account.

In the paragraph on lines 130-134 we describe a simulation scheme in which we sampled individuals from the diffusion allele frequency maps to match the locations and ages of the individuals from the Reich lab ancient DNA data and the HGDP panel.

I would recommend using latitude and longitude systematically throughout the manuscript instead of mixing x, y, latitude, longitude.

Thank you for pointing this out, in the new version of the text, we now always refer to geographic coordinates exclusively as longitude and latitude.

In the methods: Please fix the typo in Eq (9). Also please clarify which boundary conditions where used, and how the equations were solved numerically (using Eq. (8) or (9)?), including how boundary conditions were implemented.

The following explanation was added in the chapter “Parameter search” in lines 539-544.

“For numerical calculations we used the Livermore Solver for Ordinary Differential Equations (Hindmarsh, 1983) implemented in R package “deSolve” (Soetaert et al., 2010), which is a general purpose solver that can handle both stiff and nonstiff systems. In case of stiff problems the solver uses a Jacobian matrix.

Absorbing boundary conditions were used at the boundaries of the map. For visualisation purposes we masked the allele frequencies estimated in areas with negative topology (i.e. areas covered by large bodies of water).”

L437: " the left-hand side of equation 5". This doesn't look right, do you mean the right-hand side of Eq. (6)?

Thank you for pointing out the typo. The line has now been corrected.

[Editors' note: further revisions were suggested prior to acceptance, as described below.]

Reviewer #2 (Recommendations for the authors):The authors have made substantial improvements to the manuscript, and is ready for acceptance except for one remaining point:Contrary to what the authors claim in their rebuttal, Eqs (8) and (9) in Appendix 1 still contain errors.First, p(x+\δ x,y,t) and p(x+\δ x,y,t) appears twice on line 827.Second, the coefficients of p(x,y,t) on both sides of Eq (9) sum to -1, which is correct. The term should therefore be absent (have coefficient zero) in Eq (8).

Thank you for the comments and my apologies for the inconvenience caused by our previous response regarding this. We now recognize the error and have corrected it by removing the term p(x,y,t) from the equation (8).